# Revisiting flow generative models for group-wise Out-of-distribution detection

**Dihong Jiang**
School of Computer Science
University of Waterloo
Vector Institute
dihong.jiang@uwaterloo.ca

**Sun Sun**
School of Computer Science
University of Waterloo
National Research Council
sun.sun@nrc-cnrc.gc.ca

**Yaoliang Yu**
School of Computer Science
University of Waterloo
Vector Institute
yaoliang.yu@uwaterloo.ca

## Abstract

Deep generative models have been widely used in practical applications such as the detection of out-of-distribution (OOD) data. In this work, we aim to re-examine the potential of generative flow models in OOD detection. We first propose a simple combination of univariate one-sample statistical test (e.g., Kolmogorov-Smirnov) and random projections in the latent space of flow models to perform OOD detection. Then, we propose a two-sample version of our test to account for imperfect flow models. Quite distinctly, our method does not pose parametric assumptions on OOD data and is capable of exploiting any flow model. Experimentally, firstly we confirm the efficacy of our method against state-of-the-art baselines through extensive experiments on several image datasets; secondly we investigate the relationship between model accuracy (e.g., the generation quality) and the OOD detection performance, and found surprisingly that they are not always positively correlated; and thirdly we show that detection in the latent space of flow models generally outperforms detection in the sample space across various OOD datasets, hence highlighting the benefits of training a flow model.

## 1 Introduction

Reliably detecting out-of-distribution (OOD) data, also known as anomaly, plays an important role in many deployed machine learning systems, particularly so when the system may be under attack by a malicious adversary (Markou & Singh, 2003a;b; Toth & Chawla, 2018; Chalapathy & Chawla, 2019). The bulk of existing OOD detection algorithms, in one way or another, boils down to thresholding the (density) likelihood, which can be conveniently estimated by modern flow-based generative models. However, quite surprisingly, recent studies (Nalisnick et al., 2019a) reveal that flows that are trained to maximize the likelihood of in-distribution (InD) data may actually assign a higher likelihood to OOD data. For example, the flow-based model Glow (Kingma & Dhariwal, 2018) trained on CIFAR-10 that contains natural images (e.g., dog, cat, and ship) is found to assign a higher likelihood to SVHN that consists of house numbers. Such surprising and counter-intuitive observations lead to natural questions on the applicability of flow models for performing the OOD detection by merely thresholding the log-likelihood.

In this work, we re-examine the ability of flow-based generative models for OOD detection. Flow models typically transform a prior distribution that is easy to sample from to the data distribution via an invertible mapping (Tabak & Vanden-Eijnden, 2010; Rezende & Mohamed, 2015; Dinh et al., 2017; Kingma & Dhariwal, 2018). Building on this property, we propose to compare the training and test samples in the latent space of a flow model, as opposed to relying on point-wise scoring functions such as the log-likelihood. To this end, we divide test samples into minibatches (groups). Examples

of group OOD detection with generative models include for instance Nalisnick et al. (2019b); Song et al. (2019); Chalapathy et al. (2018); Zhang et al. (2020), most of which consider either the raw input or certain representation of the raw input in the sample space for detection. In contrast, we exploit data representations in the latent space, and confirm its advantage experimentally. Moreover, to cope with the curse of dimensionality (as is typical in image datasets), we propose to leverage random projections (Friedman et al., 1984), which frees one from designing any extra network architectures and is computationally very efficient. Our proposed detection algorithm combines classic univariate statistical tests and random projections in the latent space of flow models.

Besides the scoring function used for detection and the representation of data, model accuracy also greatly affects OOD detection performance (Zhang et al., 2021; Choi et al., 2018). To account for imperfectly trained flow models, we propose a two-sample version of our detection algorithm for practical use. Surprisingly, in our experiments, we confirm that model accuracy, indicated by its generation quality, may not always be positively correlated with its OOD detection performance.

We summarize our contributions as follows:

- We propose the OOD detection algorithms GOD1KS and GOD2KS for ideal and imperfect flow models, respectively, which pose no parametric assumption on the OOD data. To evade the curse of dimensionality, we propose to randomly project the latent space of flow models onto the real line and perform univariate statistical tests (such as Kolmogorov-Smirnov, KS) there. Our method is computationally very efficient, requires no extra network architecture, and unifies OOD detection in both sample space and latent space.
- Experimentally, 1) we compare with the state-of-the-art benchmarks on various image datasets and obtain competitive results; 2) we confirm on larger models and real datasets that model accuracy may not always be positively correlated with OOD detection performance; and 3) we compare detection in the sample space versus in the latent space of the flow model, and reveal the superiority and robustness of the latter, hence highlighting the potential of flow models in OOD detection.

## 2 OOD DETECTION WITH FLOW MODELS

In this section, we state the OOD detection problem and propose statistical tests that exploit modern flow generative models (either perfect or imperfectly trained).

### 2.1 OOD DETECTION: GROUP VS. POINTWISE

Let $\mathcal{D} = \{X_1, \ldots, X_n\} \overset{\text{i.i.d.}}{\sim} p$ be a sample from an in-distribution (InD) $p$. Our goal is to construct a statistical test that can decide if a test sample $\{Y_1, \ldots, Y_m\}$ is from $p$ (InD) or some unknown out-of-distribution (OOD) $q$. When $m = 1$, i.e. we examine one test sample at a time, it is often called the pointwise OOD detection (POD) while group OOD detection (GOD) refers to $m > 1$. According to Xiong et al. (2011), group OOD can be characterized as (1) point based, where each individual point in the group is anomalous, and (2) distribution based, where a group of points shows a different pattern while any single point may seem regular. Therefore, distribution based group anomaly is only detectable by GOD methods. Practical applications include detecting Higgs Bosons as a group of collision events in high-energy particle physics (Muandet & Schölkopf, 2013), and detecting distributed denial-of-services attacks via a group of multi-sensor networks (Chen & Yu, 2016). We construct a synthetic Gaussian experiment in appendix E to illustrate this point. Thus in this work, we will focus on studying the applicability of modern flow generative models to the group OOD detection problem.

### 2.2 FLOW-BASED GENERATIVE MODELS

A flow-based generative model simply learns a transformation $\mathsf{T}$, typically a diffeomorphism, that pushes a latent distribution $p_0$ (e.g., Gaussian or uniform) to the in-distribution $p$, i.e.

$$Z \sim p_0 \implies \mathsf{T}(Z) \approx p, \quad \text{also denoted as } p \approx \mathsf{T}_\# p_0. \tag{1}$$

In particular, when $\mathsf{T}$ is diffeomorphic, we have the familiar change-of-variable formula:

$$p_\mathsf{T}(\mathbf{x}) = p_0(\mathbf{z})/|\mathsf{T}'(\mathbf{z})| = p_0\left(\mathsf{T}^{-1}(\mathbf{x})\right)/\left|\mathsf{T}'\left(\mathsf{T}^{-1}(\mathbf{x})\right)\right|, \tag{2}$$

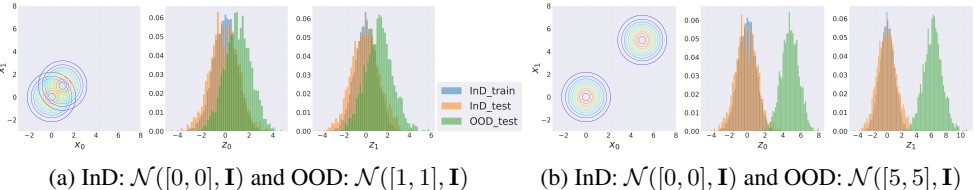

(a) InD: $\mathcal{N}([0,0],\mathbf{I})$ and OOD: $\mathcal{N}([1,1],\mathbf{I})$    (b) InD: $\mathcal{N}([0,0],\mathbf{I})$ and OOD: $\mathcal{N}([5,5],\mathbf{I})$

Figure 1: Examples of 2-D Gaussian in the latent space. In (a) and (b), left: the densities of InD and OOD; and middle and right: the density histograms of InD and OOD in each latent dimension. $x$ means input variable while $z$ denotes latent variable.

where $|\mathsf{T}'(\mathbf{z})|$ denotes the absolute value of the determinant of the Jacobian of $\mathsf{T}$. A flow model parameterizes $\mathsf{T}$ through a neural network and estimates $\mathsf{T}$ by minimizing some divergence between $p_\mathsf{T}$ (the model density) and $p$ (the data density), based on the training data $\mathcal{D}$. Once $\mathsf{T}$ is learned, we can easily generate new data by simply sampling $Z$ from $p_0$ and then pushing through $\mathsf{T}$, i.e. $X = \mathsf{T}(Z)$.

Denote $X_1 \overset{\mathrm{d}}{=} X_2$ if they follow the same distribution. For any transformation $\mathsf{T}$, it is clear that

$$X_1 \overset{\mathrm{d}}{=} X_2 \implies \mathsf{T}(X_1) \overset{\mathrm{d}}{=} \mathsf{T}(X_2), \tag{3}$$

and the converse also holds if $\mathsf{T}$ is invertible. In fact, a sufficiently regular transformation $\mathsf{T}$ may preserve many familiar statistical divergences $\mathbf{D}$, i.e.

$$\mathbf{D}(X_1, X_2) = \mathbf{D}(\mathsf{T}(X_1), \mathsf{T}(X_2)), \tag{4}$$

for instance, when $\mathbf{D}$ is the $f$-divergence, such as the Kullback-Leibler (KL) and Jensen-Shannon (JS) divergence, $\mathsf{T}$ is diffeomorphic, and $X_1$ and $X_2$ are continuous (Csiszár, 1963). Another example is the Kolmogorov-Smirnov (KS) distance for real-valued $X_i$ and monotonic $\mathsf{T}$:

$$\mathbf{D}(X_1, X_2) := \sup_x |F_1(x) - F_2(x)|, \tag{5}$$

where $F_i$ is the CDF of $X_i$, as well as the Cramér-von Mises (CvM) divergence (Darling, 1957):

$$\mathbf{D}(X_1, X_2) := \int [F_1(x) - F_2(x)]^2 \mathrm{d}F_2(x). \tag{6}$$

Thus, for group OOD detection, we can first train an invertible flow model based on the InD samples and then construct a scoring function or statistical test either in the sample space (where the test sample $\{Y_j\}$ resides), or in the latent space (where $\{\mathsf{T}^{-1}(Y_j)\}$ resides). However, since both $Y_j$ and its pre-image $\mathsf{T}^{-1}(Y_j)$ are typically of high dimension and the sample size $m$ is comparatively small, some compromise needs to be made in order to evade the curse of dimensionality, which gets even more problematic for flow models whose latent dimension is greater than input dimension (Chen et al., 2020a; Grcić et al., 2021). For instance, Zhang et al. (2020) fit a multivariate Gaussian in the latent space using $\{\mathsf{T}^{-1}(Y_j)\}$ and compute analytically the KL divergence between two Gaussians. Nalisnick et al. (2019b) project $Y_i$ onto the real line using the estimated log-likelihood function $\log p_\mathsf{T}$ and construct a typicality test there, while Choi et al. (2018) instead employ the Watanabe-Akaike Information Criterion (WAIC).

To illustrate the above idea, we compare in Figure 1 the distributions of $\mathsf{T}^{-1}(\mathbf{X}_{\text{train}})$ and $\mathsf{T}^{-1}(\mathbf{X}_{\text{test}})$ in the latent space using synthetic 2-D Gaussian datasets and the flow model RealNVP (Dinh et al., 2017). We observe that in the latent space the distribution of the OOD samples (i.e., $\mathsf{T}^{-1}(\mathbf{X}_{\text{OOD-test}})$) is distinct (more so in Fig 1(b) than Fig 1(a)) from the distribution of the InD samples (i.e., both $\mathsf{T}^{-1}(\mathbf{X}_{\text{InD-train}})$ and $\mathsf{T}^{-1}(\mathbf{X}_{\text{InD-test}})$). Moreover, we found that the transformation $\mathsf{T}^{-1}$ can roughly keep the statistical distance in the latent space. In other words, distributions that are far away from the in-distribution in the sample space also tend to remain far away in the latent space.

## 2.3 OOD DETECTION VIA RANDOM PROJECTIONS

For a perfectly trained flow model, its inverse transformation $\mathsf{T}^{-1}$ should bring the in-distribution samples to follow (approximately) the prior distribution $p_0$, such as the commonly used standard normal. It is thus natural to perform OOD detection by comparing $\mathsf{T}^{-1}(\mathbf{X}_{\text{test}})$ against the prior distribution $p_0$, whereas the InD training samples $\mathbf{X}_{\text{train}}$ are used to train the flow parameterized by

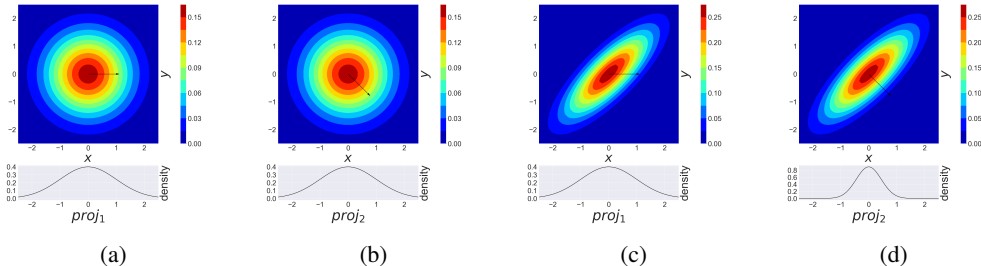

Figure 2: Random projections of 2-d Gaussians. $proj_1$ is the projection along the axis $(1, 0)$, and $proj_2$ is the projection along $(\frac{\sqrt{2}}{2}, -\frac{\sqrt{2}}{2})$. For the standard Gaussian, both $proj_1$ (a) and $proj_2$ (b) follow $\mathcal{N}(0, 1)$; while for the correlated Gaussian, $proj_1$ (c) follows $\mathcal{N}(0, 1)$, but $proj_2$ (d) follows $\mathcal{N}(0, \frac{1}{5})$, instead of $\mathcal{N}(0, 1)$.

T. To accommodate any prior distribution $p_0$, we propose to extend standard statistical tests such as the KS distance in eq. (5) to high dimensions.

To our best knowledge, multivariate KS test has only been studied sporadically in the literature, due to the computational difficulty in enumerating the maximum in high dimensions. For instance, Justel et al. (1997) proposed a complicated numerical procedure to compute the KS distance (incidentally also in the latent space) but only for bivariate distributions. Instead, we propose to randomly project high-dimensional random vectors to the real line and run classic KS test there. We note that similar ideas have been used to train generative models (e.g. Friedman et al., 1984; Bonneel et al., 2015; Kolouri et al., 2016; Liutkus et al., 2019; Paty & Cuturi, 2019; Nguyen et al., 2021), while we allow any flow models and use random projection only to construct the KS test (and related).

Figure 2 provides a simple example to illustrate the main idea. Consider two 2-d Gaussian distributions: $\mathcal{N}([0, 0], \mathbf{I})$ and $\mathcal{N}([0, 0], [1, 0.8; 0.8, 1])$. Obviously, for the standard 2-d Gaussian all of its random projections along normalized directions follow $\mathcal{N}(0, 1)$, whereas for the shown correlated Gaussian, only the projections along the coordinate axes follow $\mathcal{N}(0, 1)$. Therefore, we can distinguish these two distributions by comparing their projections along random directions.

More generally, the following theorem provides the theoretical basis for distributional comparison using random projections.

**Theorem 1** (Cuesta-Albertos et al. 2007). *Let $X$ and $Y$ be two $\mathbb{R}^d$-valued random vectors. Suppose the absolute moments $m_k := \mathbb{E}\|X\|^k$ are finite and $\sum_{k=1}^{\infty} (m_k)^{-1/k} = \infty$. If the set $W = \{\mathbf{w} \in \mathbb{R}^d : \mathbf{w}^\top X \stackrel{\mathrm{d}}{=} \mathbf{w}^\top Y\}$ has positive Lebesgue measure, then $X \stackrel{\mathrm{d}}{=} Y$.*

The assumption $\sum_k (m_k)^{-1/k} = \infty$, known as Carleman's condition, is very mild: it is satisfied if the underlying moment generating function is finite around the origin (hence such distributions are uniquely determined by their moments). Most distributions used in practice, such as the Gaussian distribution, clearly satisfy Carleman's condition. Put differently, Theorem 1 implies that a single random direction almost surely allows us to distinguish the projections of $X$ and $Y$; see Figure 2 for an illustration. We note that Theorem 1 can be slightly strengthened if we project to higher dimensional subspaces (Cuesta-Albertos et al., 2007), which, however, renders the classic KS test inapplicable. Therefore, in this work, we will only consider random projections onto the real line.

To be more specific, given a limited collection of test samples $\mathbf{X}_{\text{test}}$ and a flow model T, we first transform into the latent space and obtain $\mathbf{Z} = \mathsf{T}^{-1}(\mathbf{X}_{\text{test}})$. Then, we sample $n$ (uniformly) random directions $\mathbf{W} \in \mathbb{R}^{d \times n}$, which are obtained by normalizing i.i.d. samples from the $d$-dimensional standard Gaussian. When the flow model T is well-trained, the inverse transformation $\mathsf{T}^{-1}$ applied to InD samples will bring them to follow approximately the latent prior distribution, e.g., $d$-dimensional standard Gaussian. As a result, projection onto each random direction yields close proximity to the univariate standard Gaussian, which classic statistical tests such as KS would be able to pick up. In practice, we found that averaging over different random directions leads to slightly better and more robust performance, although the benefits quickly saturate as we increase the number of random projections. Crucially, the KS distance in eq. (5), with $F_1$ being the empirical

---

**Algorithm 1:** Group OOD detection based on one-sample KS test (GOD1KS).

---

**Input:** Test OOD samples $\mathbf{X}_{\text{test}}$ splitted into $m$ groups $\mathbf{X}_1, \cdots, \mathbf{X}_m$ with each $\mathbf{X}_i \in \mathbb{R}^{d \times b}$ ($b$ for batch size and $d$ for dimension); random projection matrix $\mathbf{W} \in \mathbb{R}^{d \times n}$.

1 **for** $i \leftarrow 1$ *to* $m$ **do**
2 $\quad$ $\mathbf{Z}_i = \mathsf{T}^{-1}(\mathbf{X}_i) \in \mathbb{R}^{d \times b}$ $\qquad\qquad$ // transform into the latent space
3 $\quad$ $\mathbf{S}^{(i)} = \mathbf{W}^\top \mathbf{Z}_i \in \mathbb{R}^{n \times b}$ $\qquad\qquad$ // project onto $n$ random directions
4 $\quad$ **for** $j \leftarrow 1$ *to* $n$ **do**
5 $\quad\quad$ $k_{ij} = \mathrm{KS}(\mathbf{S}^{(i)}_{j:}, \mathcal{N}(0,1))$ $\qquad\qquad$ // conduct one-sample KS test
6 $\quad$ $k_i \leftarrow \frac{1}{n} \sum_{j=1}^n k_{ij}$ $\qquad\qquad$ // average over $n$ random directions
7 compute AUROC for $\mathbf{X}_{\text{test}}$ based on all $k_i$'s

---

distribution of random projections $W^\top \mathbf{Z}$ and $F_2$ the latent prior distribution, can be computed in linear time by just enumerating $x$ over each projected sample.

We call the resulting algorithm group OOD detection based on one-sample KS test (GOD1KS) and summarize it in Algorithm 1. Its computation requires one pass of the (inverse) flow model and the remaining operations are linear-time. A higher value of the KS statistics $k_{ij}$ indicates a lower similarity between the test sample and the latent prior distribution, which can be taken as a metric of OOD-ness. We note that our algorithm is completely general and efficient:

- unlike Zhang et al. (2020) we do not require any matrix inversion or determinant and we avoid the difficult problem of estimating high dimensional covariance matrices when only very limited test samples are available;

- in principle, we can work with any latent prior distribution and any univariate statistical tests. As pointed out by Jaini et al. (2020), a heavier tailed latent distribution, or even discrete ones, than the standard Gaussian may be advantageous in certain settings. Similarly, other statistical tests may prove useful if we desire to zoom in certain parts of the distribution. Our choice of the KS test is motivated by our experimental settings below and serves as a concrete example.

- our algorithm can work with any flow model $\mathsf{T}$. For instance, we may even take $\mathsf{T} = \mathrm{Id}$, in which case Algorithm 1 reduces to performing statistical tests in the sample space. Thus, our algorithm unifies the two perspectives: test in the sample space vs. test in the latent space, which we will compare experimentally below.

### 2.4 IMPROVEMENT FOR IMPERFECT FLOW MODELS

When the flow model is not well-trained (perhaps even intentionally, for instance, if we take $\mathsf{T} = \mathrm{Id}$), the effectiveness of the one-sample test in Algorithm 1 becomes questionable even for InD samples. A simple fix is then to run the two-sample version of the KS test in eq. (5), where $F_1$ is the empirical distribution of the projected test samples while $F_2$ is now the empirical distribution of the InD training samples. More concretely, in Algorithm 1 we additionally derive $\mathbf{Z}_{\text{train}} = \mathsf{T}^{-1}(\mathbf{X}_{\text{train}})$ and project similarly to obtain $\mathbf{S}_{\text{train}} = \mathbf{W}^\top \mathbf{Z}_{\text{train}}$. Then, in Line 5 we substitute the latent prior distribution (e.g. standard Gaussian) with the empirical distribution of the $j$-the row of $\mathbf{S}_{\text{train}}$. Below we call this modification as group OOD detection based on two-sample KS test (GOD2KS). We note that the computational complexity of GOD2KS remains similar to Algorithm 1, and it is equally flexible: we can now even take snapshots of $\mathsf{T}$ obtained during training the flow model, and run GOD2KS on all of them. The detailed algorithm for GOD2KS is presented in Appendix B.

## 3 EXPERIMENTAL RESULTS

We perform extensive experiments to compare our proposed OOD detection algorithms with the state-of-the-art (SOTA) group OOD detection benchmarks, i.e. Typicality test (TyTest) (Nalisnick et al., 2019b) and the KL divergence based Out-of-Distribution Detection (KLOD) (Zhang et al., 2020). Implementation details of benchmarks are given in Appendix C. We test on two popular flow models: Glow (Kingma & Dhariwal, 2018) and RealNVP (Dinh et al., 2017) (see Appendix B

Table 1: AUROC on RealNVP (higher is better). Our results are denoted by GOD1KS|GOD2KS. Highest AUROC are in boldface, and failure cases (where AUROC is below 0.5) are underlined.

| InD | OOD | batch size = 5 | | | batch size = 10 | | |
|---|---|---|---|---|---|---|---|
| | | TyTest | KLOD | Ours | TyTest | KLOD | Ours |
| FMNIST | MNIST | 0.77 | 0.95 | **0.99**\|**0.99** | 0.82 | 0.96 | **1.00**\|**1.00** |
| | KMNIST | 0.97 | 0.97 | **0.99**\|**0.99** | 0.97 | 0.98 | **1.00**\|**1.00** |
| | Omniglot | 1.00 | 1.00 | 1.00\|1.00 | 1.00 | 1.00 | 1.00\|1.00 |
| CIFAR-10 | SVHN | **0.89** | 0.29 | 0.88\|0.83 | 0.95 | 0.35 | **0.99**\|0.98 |
| | CelebA | 0.67 | **0.90** | 0.76\|0.77 | 0.81 | **0.99** | 0.89\|0.90 |
| | LSUN | 0.60 | **0.72** | 0.61\|0.62 | 0.70 | **0.81** | 0.66\|0.70 |
| CIFAR-100 | SVHN | 0.92 | 0.23 | 0.89\|0.84 | 0.98 | 0.29 | **0.99**\|0.98 |
| | CelebA | 0.42 | **0.83** | 0.74\|0.74 | 0.46 | **0.97** | 0.88\|0.88 |
| | LSUN | 0.49 | **0.63** | 0.57\|0.59 | 0.53 | **0.72** | 0.61\|0.65 |
| SVHN | CIFAR-10 | **1.00** | 0.99 | 0.89\|0.93 | **1.00** | 0.99 | 0.95\|0.98 |
| | CIFAR-100 | **1.00** | 0.99 | 0.90\|0.93 | **1.00** | 1.00 | 0.96\|0.98 |
| | CelebA | **1.00** | **1.00** | 0.92\|0.94 | **1.00** | **1.00** | 0.98\|0.99 |
| | LSUN | **1.00** | **1.00** | 0.93\|0.94 | **1.00** | **1.00** | 0.98\|0.99 |
| CelebA | CIFAR-10 | 0.98 | **0.99** | 0.92\|0.93 | **1.00** | **1.00** | 0.98\|0.99 |
| | CIFAR-100 | 0.98 | **0.99** | 0.91\|0.92 | **1.00** | **1.00** | 0.98\|0.99 |
| | SVHN | 0.78 | 0.80 | **0.97**\|0.96 | 0.81 | 0.99 | **1.00**\|**1.00** |
| | LSUN | **1.00** | **1.00** | 0.91\|0.93 | **1.00** | **1.00** | 0.98\|0.99 |

for more details about network architecture). For evaluation, we focus on the Area Under Receiver Operating Characteristic (AUROC/AUC) and Area Under Precison-Recall curve (AUPR), which are commonly used in OOD detection. We compare the OOD detection performance across a wide variety of image datasets, including grayscale and RGB image datasets (please refer to Appendix A for more details).

## 3.1 ROBUSTNESS OF GOD2KS

In Table 1 we compare the performance of GOD1KS and GOD2KS with the flow model RealNVP against the benchmark algorithms. For fair comparison, we adopt the same RealNVP model in the benchmarks. AUPR comparison is given in Table 5 in appendix G.2. We use InD to denote the in-distribution dataset and OOD to denote different test datasets. To implement group OOD detection, we divide the test samples into small batches and vary the batch size for comparison. We use 50 random projections in all cases, and set it as the default for all experiments on RealNVP. As expected, the performance improves with the batch size (see Appendix G.5 for results with batch size 20). We observe that, the performance of the two benchmarks and our GOD1KS on these image datasets is unstable, and for some datasets the detection performance can be rather poor (i.e., AUROC below 0.5). For example, for TyTest with CIFAR-100 (InD) and CelebA (OOD), the AUROCs are 0.42 and 0.46; for KLOD with CIFAR-10 (InD) and SVHN (OOD), the AUROCs are 0.29 and 0.35. In contrast, the performance of our GOD2KS is more robust and is generally satisfactory over all datasets. Results on Glow is shown in Table 6 in Appendix G.3. Again, our GOD2KS exhibits robustness over all datasets. Benchmarks also discuss detecting CIFAR-100 as OOD when training a model on CIFAR-10, and their AUROCs are around random guess. We did the same experiment and obtain similar results. The result is not surprising, since CIFAR-10 and CIFAR-100 are similar datasets, which highlights the need to quantify OOD-ness.

## 3.2 RANDOM PROJECTION VS. AUTOENCODERS

When the number of random projections is less than the input dimension, we essentially perform dimensionality reduction before comparing the distributions. Therefore, it is natural to compare with other dimensionality reduction methods such as autoencoders. To construct the benchmark, we first feed the input image to an autoencoder and then use the latent code from the encoder as the input in our GOD1KS/GOD2KS. The only difference is that with autoencoder we now skip the step of random projections. As an example, we use the Latent Space Autoregression (LSA) (Abati et al., 2019) as the autoencoder. Figure 3 shows the comparison results on FMNIST/MNIST (InD/OOD) and CIFAR-10/SVHN using RealNVP. The batch size is fixed as 10. We vary the number of random

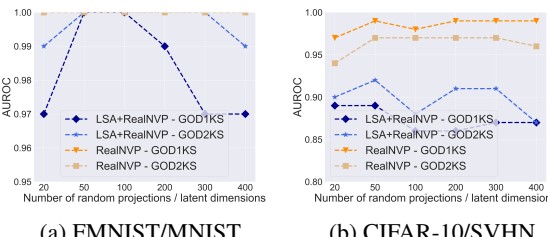

(a) FMNIST/MNIST     (b) CIFAR-10/SVHN

Figure 3: Random projections vs. autoencoder. (a)-(b) Performing random projection outperforms training an extra autoencoder in two OOD detection experiments. Besides, the OOD detection performance using RealNVP saturates as the number of random projections grows beyond 50.

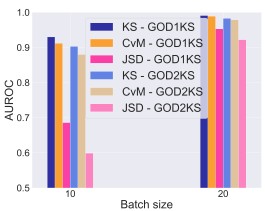

Figure 4: Different divergences. KS and CvM test outperform JSD. A further comparison between KS and CvM is presented in appendix G.4.

projections or the latent dimension from 20 to 400, and found that using random projections outperforms autoencoder with the same level of dimensionality reduction. It is possible to improve the detection performance with an autoencoder by using a more complex network than LSA. However, random projection is still appealing, as it works effectively and requires no extra networks.

## 3.3 Effect of divergence measures

We empirically compare the KS test with the CvM test and the JS divergence (JSD). Both the KS and the CvM tests are non-parametric, while JSD requires the density to be estimated. We use the following heuristic approach to estimate the empirical density: dividing the latent values of the test batch into 20 bins, and use the normalized count in each bin as the empirical density (note that we can skip this step in both KS and CvM). Therefore, the computation time for calculating JSD is much longer. For implementation, we use the Scipy.stats library in Python to computation these measures. As an example, in Figure 4 we show results on one challenging dataset pair CIFAR-10/SVHN (InD/OOD) using RealNVP (results on Glow are similar). We consider two batch sizes: 10 and 20. The number of random projections is set to be 10 for all experiments. We can observe that JSD is outperformed by the other two in all cases, and the performance of KS and CvM is comparable. We further compare KS and CvM on more dataset pairs for different batch sizes (see Appendix G.4). The observation is generally consistent with KS being slightly superior to CvM, especially for the smaller batch size.

## 3.4 Generation quality vs. OOD detection

Intuitively, a more accurate flow model is expected to lead to better OOD detection performance. In this section, we investigate the relationship between the model accuracy and the OOD detection performance, where we measure a model's accuracy by its ability to generate visually realistic high-quality new images. To impose different levels of model accuracy we consider two factors: model capacity and training time. We fix the batch size to 10 in this experiment.

**Model capacity:** Consider Glow, where $K$ denotes the number of steps of flow in each block, $L$ denotes the number of blocks, and $h$ denotes the number of hidden channels. For comparison we trained two different Glow models on CIFAR-10: (1) a simple Glow with $K = 3, L = 3, h = 64$, and (2) a complex Glow with $K = 16, L = 3, h = 128$. Figure 5 shows the comparison of the log-likelihood histograms and the generated images. We can see that while the histograms of the two log-likelihood are similar, the generation quality is noticeably different with the complex model generating much better images. We then run our OOD detection algorithms with SVHN as OOD. Surprisingly, we found that the simpler Glow yields better OOD detection performance. We also observed similar results with RealNVP (see Appendix G.6 for details).

**Training time:** In appendix G.6, we visualize how the generation quality and the OOD detection performance evolve with training time. We use RealNVP with 16 blocks and 512 hidden channels trained on CelebA. Generally, more training time leads to a better model, which is indicated

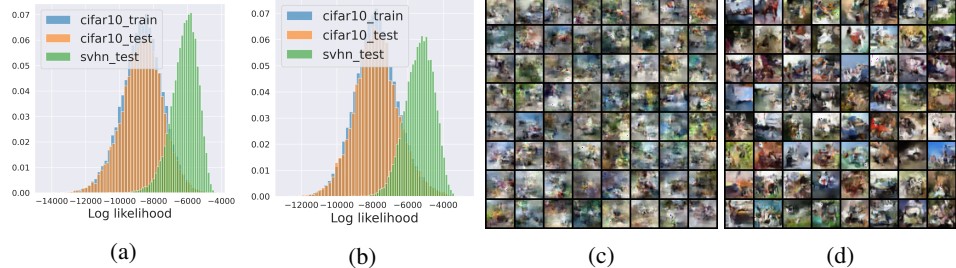

(a)          (b)          (c)          (d)

Figure 5: Model capacity vs. OOD detection. While the density histograms estimated from the simple Glow (a) and complex Glow (b) are similar, the generation from the simple Glow (c) and complex Glow (d) are noticeably different. More surprisingly, the OOD detection performance by simple Glow (AUROCs: 0.99|0.99) is better than the complex Glow (AUROCs: 0.96|0.95).

by the generation quality. Taking CIFAR-10 as OOD, we observe that the detection performance improves with training time, as shown in Figure 9. However, while we have similar observations on most dataset pairs, we still found some anomalies, e.g., Glow on CIFAR-10/SVHN and FM-NIST/MNIST, where the detection performance fluctuates or even declines with the training time (see Appendix G.6).

We conclude that model accuracy, as measured by the generation quality in our experiments, is not always positively correlated with the OOD detection performance. These results confirm the findings of Zhang et al. (2021); Schirrmeister et al. (2020) on more models and more real datasets, and reveal the surprising phenomenon that a misestimated model can sometimes lead to better OOD detection. We present our detailed explanation and discussion in Appendix G.6.

### 3.5 DISTRIBUTION COMPARISON IN LATENT VS. SAMPLE SPACES

Instead of transforming the data into the latent space, we can directly apply our detection method to the raw inputs without training a flow model, i.e. set $\mathsf{T} = \mathrm{Id}$. Naturally, one wonders if it is necessary to use a flow model for OOD detection. Or, can we detect in the sample space directly? Thanks to the generality of our algorithm, we are positioned to report a fair comparison below.

When performing detection in the sample space directly with our method, we can only use the two-sample version due to the absence of the prior distribution. In this experiment, the batch size is fixed to be 10 and the number of random projections is fixed to be 50. We first run our GOD2KS (without the inverse transformation) in the sample space on image datasets, and compare with the detection results obtained in the latent space, as shown in Table 3 in appendix G.1. Interestingly, we found that detecting in the latent space is generally better than in the sample space, especially for SVHN/CIFAR-10, SVHN/CIFAR-100, and SVHN/CelebA.

Next, we examine the detection performance by varying the OOD distribution. To this end, we manually inject zero-mean Gaussian noise to the OOD data. The results are summarized in Table 4 in appendix G.1. We can see that the detection performance in the sample space varies with the OOD distribution, while that in the latent space is superior and stable. We hypothesize that the robustness of detection in the latent space can be attributed to the more regular structure of the latent prior distribution (i.e., Gaussian), which can be beneficial for distributional comparison.

## 4 RELATED WORK

OOD detection has been explored from different perspectives, e.g., discriminative methods, generative models, or hypothesis tests; see Toth & Chawla (2018); Chalapathy & Chawla (2019); Pang et al. (2021) for extensive reviews. Below we only discuss papers on unsupervised methods using generative models.

**Unsupervised group OOD detection** Nalisnick et al. (2019b) propose the typicality test for group OOD detection under the hypothesis that the in-distribution samples are drawn from the typical set of the data distribution, which may not overlap with the regions of high density. This hypothesis is

recently interrogated by Zhang et al. (2021), who instead attribute the failure of OOD detection in deep generative models to model misestimation. Moreover, Song et al. (2019) observe that batch normalization can lower the likelihood of a batch of OOD samples, based on which a permutation test is proposed for group OOD detection. Chalapathy et al. (2018) define a group reference function which aggregates the information of input groups and then suggests a distance score for OOD detection that measures the deviation between a test group and the group reference function. It's worth mentioning that both Song et al. (2019) and Chalapathy et al. (2018) adopt a fixed and large batch size (64 and 1536, respectively) for all experiments, and thus it is not clear how the batch size (especially the smaller ones) would affect the detection performance. Our work is most related to Zhang et al. (2020), who also consider detection in the latent space of flow models. The main differences lie in three aspects: (1) they impose the Gaussian assumption on the latent distribution of test samples, so that they can compare it with the latent prior in a closed form using KL-divergence. In contrast, our KS test is non-parametric, hence does not require any distributional assumption on test data; (2) they assume the trained flow model is perfect, while we additionally provide GOD2KS for imperfect models; and (3) we use random projections to consider both marginal distribution and correlations of all dimensions, while they need to explicitly estimate the correlation coefficients to quantify the inter-dimensional correlations, which can be challenging in practice.

**Unsupervised point OOD detection**  Compared to group OOD detection, point OOD detection treats test samples individually. In the context of deep generative models, many researchers have tried to explain the failure of log-likelihood for OOD detection from different perspectives, e.g., the background statistics (Ren et al., 2019), the inductive biases (Kirichenko et al., 2020), the input complexity (Serrà et al., 2020), and the model accuracy (Choi et al., 2018), and proposed new metrics therein. Our work can also be extended to point detection by artificially creating a batch of samples for each individual test sample (see Appendix F for more details). Note that Schirrmeister et al. (2020); Havtorn et al. (2021) also develop detection scores based on (partial) latent variables of flow model and VAE, respectively. Differently, Havtorn et al. (2021) interpret their likelihood ratio test as comparing divergence between true and approximate posterior in latent space (since true posterior is intractable), whereas Schirrmeister et al. (2020) contrast the log likelihood of the last-scale latent code that encodes global object-specific features with that of some background latent code. Nevertheless, both approaches do not employ random projections or KS tests as we do. Morningstar et al. (2021) extended the typicality test to the pointwise setting, by fitting a kernel density estimator to each of a few test statistics on the training set and then computing the sum of log probability of each statistic as the detection score for a given data point.

**Representation learning for OOD detection**  OOD detection is challenging in high dimensions. This motivates the study of representation learning, which aims to characterize discriminative information of the input with a reduced dimension for the OOD detection. Example methods include projection (Pevnỳ, 2016; Pang et al., 2018), feature selection (Azmandian et al., 2012; Pang et al., 2017), and deep learning based methods. For the latter one, representations can be obtained either from a pre-trained model (Zhou et al., 2019; Pang et al., 2020; Andrews et al., 2016; Tudor Ionescu et al., 2017), or be directly learned with a neural network such as autoencoders (Xu et al., 2015; Ionescu et al., 2019; Erfani et al., 2016; Wang et al., 2019). In our work, the representation is obtained by random projecting the latent space of flow models. Compared with other deep learning based approaches, our method requires no additional networks and is computationally more efficient.

## 5  CONCLUSION

In this work, we re-examined the potential of flow models for OOD detection. We provided practical OOD detection algorithms that compare distributional information in the latent space and impose no parametric assumption. We compared with SOTA benchmarks and obtained comparable and more stable performances. Experimentally, we demonstrated the benefits of detecting OOD data in the latent space and confirmed that OOD detection performance is not always positively correlated with model accuracy. In the future, we would like to characterize OOD distributions that can be reliably detected by our methods, as well as the possibility of projecting onto high dimensional subspaces.

## ACKNOWLEDGMENTS

We thank the reviewers as well as the area chair for thoughtful comments. We gratefully acknowledge funding support from NSERC, the Canada CIFAR AI Chairs program, NRC and University of Waterloo. Resources used in preparing this research were provided, in part, by the Province of Ontario, the Government of Canada through CIFAR, and companies sponsoring the Vector Institute.

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

# A  DATASETS

In this section, we briefly introduce the public datasets we use and our preprocessing in this work. Note that image size is represented in $\#channels \times height \times width$, and all images are normalized into $[0.0, 1.0]$.

- Grayscale image datasets:
    - MNIST (LeCun et al., 1998): MNIST is a dataset of handwritten digits, including 10 classes (from digit 0 to 9) and 70000 images in total. Each image is in $1 \times 28 \times 28$.
    - FMNIST (Xiao et al., 2017): FMNIST is a dataset of Zalando's article images with 10 classes of clothes and shoes. There are 70000 images in total, and each of them is in $1 \times 28 \times 28$.
    - KMNIST (Clanuwat et al., 2018): KMNIST is a dataset of Japanese characters. There are 10 character labels and 70000 images in total, and each of them is in $1 \times 28 \times 28$.
    - Omniglot (Lake et al., 2015): Omniglot is a dataset of handwritten digits from different alphabets. There are 1623 different characters and 32460 images in total, and each of them is in $1 \times 105 \times 105$. We center-cropped images to the size of $1 \times 84 \times 84$, and then resize them into $1 \times 28 \times 28$.
- RGB image datasets:
    - CIFAR-10/100 (Krizhevsky et al., 2009): CIFAR-10/100 are datasets of natural images, e.g. animals and vehicles. CIFAR-10 contains 10 classes, while CIFAR-100 contains 100 classes. Both datasets contain a total of 60000 images. Each image is in $3 \times 32 \times 32$.
    - SVHN (Netzer et al., 2011): SVHN is a dataset of street view housing numbers, including 10 classes and 99289 images in total. Each image is in $3 \times 32 \times 32$.
    - LSUN (Yu et al., 2015): LSUN is a dataset of 10 scene categories, e.g., bedroom, classroom, kitchen. Only 10000 test images are used in this work. Images are not in an universal size. The smaller dimension is resized to 256 by the original dataset. We center-cropped images to the size of $3 \times 224 \times 224$, and then resize them into $3 \times 32 \times 32$.
    - CelebA (Liu et al., 2015): CelebA is a dataset of face images of celebrities. It contains a total of 10177 identities and 202599 face images. Each image is in $3 \times 178 \times 218$. We center-cropped images to the size of $3 \times 178 \times 178$, and then resize them into $3 \times 32 \times 32$. 50000 images are randomly sampled from predefined training split as the trainig set in this work.

We use the official training and test split for all datasets, and we create the validation set by randomly holding out 10% from the training split.

# B  IMPLEMENTATION DETAILS ON OUR METHOD

- Glow: Our Pytorch implementation of Glow was derived from Joost van Amersfoort's repository[1]. In our main results, for grayscale image datasets, we use 2 blocks, where each block consists of 3 steps of flow with 64 hidden channels. For RGB image datsets, we use 3 blocks, where each block consists of 3 steps of flow with 64 hidden channels. Affine coupling layers with squeezing factor of 2 are used for all datasets. All priors are set as standard Gaussians. The learning rate is set as 1e-5 for grayscale image datasets and 1e-3 for RGB image datasets. The optimizer is Adam with a weight decay of 1e-6.
- RealNVP: Our Pytorch implementation of RealNVP was derived from Ilya Kostrikov's repository[2]. In our main results, for grayscale image datasets, we use one block with 512 hidden channels. For RGB image datsets, we use 1 block with 2048 hidden channels. All priors are set as standard Gaussians. The learning rate is set as 1e-6 for grayscale image datasets and 1e-5 for RGB image datasets. The optimizer is Adam with a weight decay of 1e-6.

As mentioned earlier, the training epochs can significantly affect the OOD detection performance, and we found that the two SOTA baselines, as well as our GOD1KS, are more sensitive to training time than GOD2KS. We use validation set to select the training epoch that optimizes our GOD2KS

---

[1] https://github.com/y0ast/Glow-PyTorch
[2] https://github.com/ikostrikov/pytorch-flows

---

**Algorithm 2:** Group OOD detection based on two-sample KS test (GOD2KS).

---

**Input:** Training InD samples $\mathbf{X}_{\text{train}} \in \mathbb{R}^{d \times N}$ ($N$ denotes the size of training set), Test OOD samples $\mathbf{X}_{\text{test}}$ splitted into $m$ groups $\mathbf{X}_1, \cdots, \mathbf{X}_m$ with each $\mathbf{X}_i \in \mathbb{R}^{d \times b}$ ($b$ for batch size and $d$ for dimension); random projection matrix $\mathbf{W} \in \mathbb{R}^{d \times n}$.

1   $\mathbf{Z}_{\text{train}} = \mathsf{T}^{-1}(\mathbf{X}_{\text{train}}) \in \mathbb{R}^{d \times N}$

2   $\mathbf{S}_{\text{train}} = \mathbf{W}^\top \mathbf{Z}_{\text{train}} \in \mathbb{R}^{n \times N}$

3   **for** $i \leftarrow 1$ **to** $m$ **do**

4     $\mathbf{Z}_i = \mathsf{T}^{-1}(\mathbf{X}_i) \in \mathbb{R}^{d \times b}$          `// transform into the latent space`

5     $\mathbf{S}^{(i)} = \mathbf{W}^\top \mathbf{Z}_i \in \mathbb{R}^{n \times b}$         `// project onto` $n$ `random directions`

6     **for** $j \leftarrow 1$ **to** $n$ **do**

7       $k_{ij} = \text{KS}(\mathbf{S}^{(i)}_{j:}, \mathbf{S}_{\text{train}(j:)})$       `// conduct two-sample KS test`

8     $k_i \leftarrow \frac{1}{n} \sum_{j=1}^n k_{ij}$          `// average over` $n$ `random directions`

9   compute AUROC for $\mathbf{X}_{\text{test}}$ based on all $k_i$'s

---

performance. Specifically, for RealNVP, we found that training the model for 50 epochs on FMNIST, CIFAR-100, CelebA, and 100 epochs on CIFAR-10, SVHN, achieves better GOD2KS results. For Glow, we found that training the model for 2 epochs on CIFAR-10, 5 epochs on CIFAR-100, and 200 epochs on FMNIST, SVHN, CelebA, achieves better GOD2KS results.

We also present the detailed algorithm of our GOD2KS for completeness, as shown in Algorithm 2.

## C   Implementation details of benchmarks

For TyTest, we follow their Algorithm 1 except the threshold $\epsilon$ to calculate the detection score, since we are using AUROC for comparison. For KLOD, we follow their equation (4) to calculate the detection score. However, we observe one empirical issue of KLOD, i.e. the estimation of a high-dimensional covariance matrix from a small batch are mostly singular, which leads the log determinant of covariance matrix in equation (4) of Zhang et al. (2020) to be negative infinity, and thus not indicative as a detection score at all. In our experiment, we calculate $\mu^\top \mu$ to replicate their results.

## D   Framework

Our framework is illustrated in Figure 6. When the flow model $\mathsf{T}^{-1}$ is trained, we do the inference to obtain the latent variable, i.e. $z = \mathsf{T}^{-1}(x)$, then do the detection based on the distribution of $z$.

## E   Motivation for groupwise detection

As mentioned in section 2.1, distribution based group anomaly is only detectable by groupwise detection methods. We conduct a synthetic experiment to illustrate this point. Consider two 2-d Gaussians with the same zero mean and orthogonal eigenvectors, e.g. InD=$\mathcal{N}([0, 0], [[1, 0.5], [0.5, 1]])$ and OOD=$\mathcal{N}([0, 0], [[1, -0.5], [-0.5, 1]])$. The two distributions have a large overlap around origin where the majority of mass reside in (they have the same density peak). Therefore, if we train a generative model (e.g. RealNVP in this example) to estimate their density, the InD and OOD density histograms will largely overlap, as shown in Figure 7, which results in an AUROC of pointwise likelihood detection score as low as 0.57. In contrast, with increasing number of test samples drawn from these two distributions, the separation of InD and OOD will be increasingly clear (which is true for both sample space and latent space). Our GOD1KS and GOD2KS methods achieve detection AUROC of 0.67|0.67 for batch size = 10, 0.77|0.77 for batch size = 20, and 0.97|0.96 for batch size = 100, respectively.

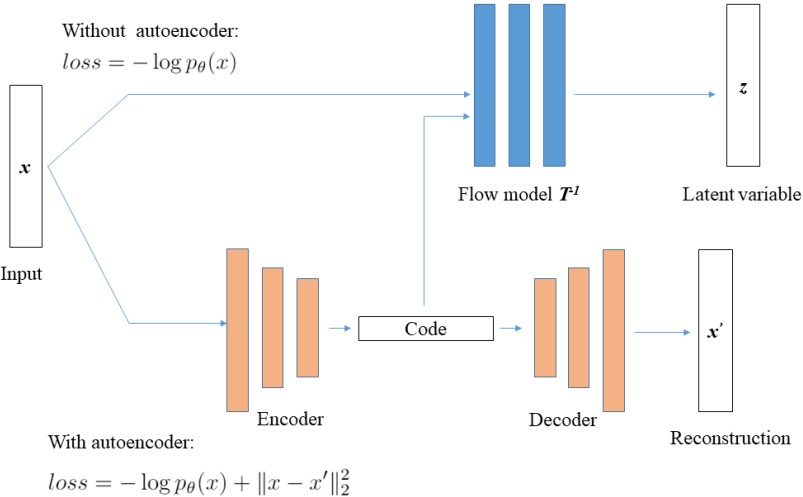

Figure 6: The framework of our method.

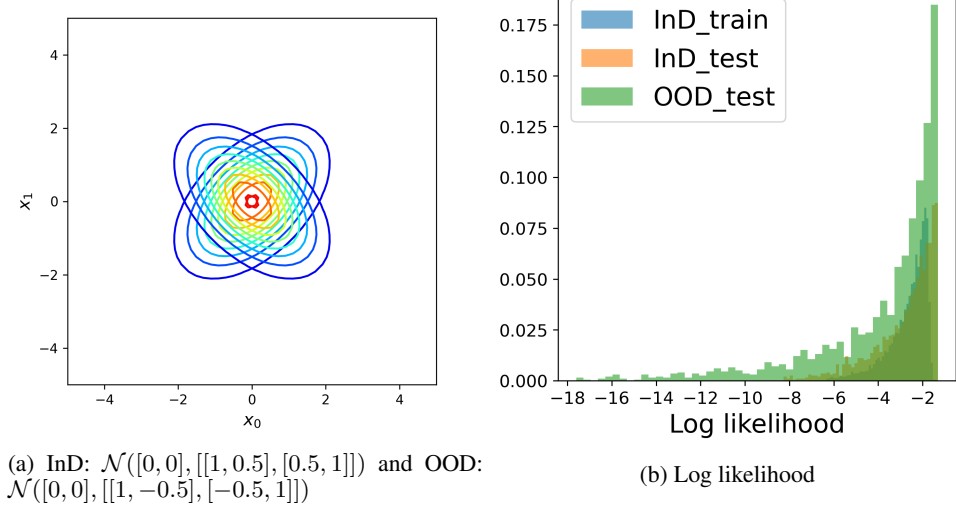

(a) InD: $\mathcal{N}([0,0], [[1, 0.5], [0.5, 1]])$ and OOD: $\mathcal{N}([0,0], [[1, -0.5], [-0.5, 1]])$

(b) Log likelihood

Figure 7: An example of distribution based group anomaly. (a) Two 2-d Gaussians with similar density and large overlap in support. (b) Density histograms largely overlap, where likelihood score only achieves a detection AUROC of 0.57.

## F  POINT OOD DETECTION VIA DATA AUGMENTATION

In GOD, we assume that a group of test samples are either from the in-distribution or not. Now we will relax this assumption and extend the proposed group OOD detection to point OOD detection.

The key of the extension is to obtain a group of samples based on one single test sample. To do so, one option is to adopt data augmentation (Shorten & Khoshgoftaar, 2019), which has been extensively used in deep learning when the number of the training samples is limited. Data augmentation is also a common practice to construct the positive pairs in contrastive learning (Chen et al., 2020b). Concretely, for each image sample we can construct a group of samples by applying, e.g., rotations, random cropping and resizing, and color distortion. For comparison, we produce 10 and 40

samples respectively. Specifically, 10 augmented samples consist of 3 rotated images (90, 180, 270 degrees), 3 random cropped and resized images, and 3 color distorted (jitter) images, as well as the original one, while 40 augmented samples consist of 3 rotated images (90, 180, 270 degrees), 18 random cropped and resized images, and 18 color distorted (jitter) images, as well as the original one. Generally, our detection performance improves with the number of data augmentation.

We compare the performance of our point OOD algorithm with two SOTA benchmarks: $\log q(\mathbf{x})$ and input complexity $\mathcal{S}$ (Serrà et al., 2020), as shown in Table 2. Both $\log q(\mathbf{x})$ and $\mathcal{S}$ encounter complete failure detection case, e.g. CIFAR-10/SVHN for $\log q(\mathbf{x})$ and FMNIST/KMNIST for $\mathcal{S}$, while our POD is more robust to different dataset pairs.

Table 2: AUROC on RealNVP for point OOD detection (higher is better). Our results are denoted by R1|R2, where R1 corresponds to GOD1KS, and R2 corresponds to GOD2KS.

| InD | OOD | $\log q(\mathbf{x})$ | $\mathcal{S}$ | Ours-10 | Ours-40 |
|---|---|---|---|---|---|
| FMNIST | MNIST | 0.80 | 0.43 | 0.74\|0.75 | 0.79\|0.81 |
| | KMNIST | 0.91 | 0.19 | 0.84\|0.83 | 0.90\|0.89 |
| | Omniglot | 1.00 | 1.00 | 1.00\|1.00 | 1.00\|1.00 |
| CIFAR10 | SVHN | 0.16 | 0.80 | 0.85\|0.81 | 0.85\|0.82 |
| | CelebA | 0.65 | 0.51 | 0.59\|0.59 | 0.57\|0.57 |
| | LSUN | 0.60 | 0.48 | 0.45\|0.47 | 0.46\|0.48 |

## G  ADDITIONAL RESULTS

### G.1  SAMPLE SPACE VS. LATENT SPACE

Experimental results of detecting in sample space and latent space are summarized in table 3 and table 4 (with injecting noise to OOD data).

### G.2  GOD RESULTS USING REALNVP - AUPR

The corresponding AUPRs for the experiments in Table 1 are shown in Table 5.

### G.3  GOD RESULTS USING GLOW

The optimal number of random projections can be dataset- and model-dependent. We run a similar experiment as section 3.2 with Glow to select the optimal number of projections. The selection process is similar to section 3.2, thus is omitted. Finally, we use 100 random projections when training Glow on grayscale image datasets, and use 200 random projections when training Glow on RGB image datasets. The detection result is summarized in Table 6. Comparing Table 6 with Table 1, we can notice that the detection performance can be largely affected by the network architecture. In particular, for KLOD and our GOD1KS the AUROC for FMNIST (InD) and MNIST/KMNIST (OOD) can be close to 0, which indicates a complete failure of detection. The underlying reason is that it is always questionable that a trained flow model is 'perfect', which means there is always possibility that $T^{-1}_{\#\text{OOD}}$ is closer to the latent prior than $T^{-1}_{\#\text{InD}}$, motivating us to propose the two-sample version to improve the robustness.

### G.4  COMPARISON OF DIFFERENT DIVERGENCE MEASURES

We conduct a few more experiments to compare KS test and CvM test, as shown in Table 7, where the KS test indicates more robustness especially when batch size is small.

### G.5  BATCH SIZE = 20

We add more results when batch size is 20, as shown in Table 8. The model is the same as Table 1 and 6. As expected, the GOD performance will increase as batch size increases. Our method can achieve an AUROC of 1.00 for most cases when batch size is 20.

Table 3: Comparison in the sample and latent spaces. AUROCs are shown and segmented by |. S denotes detection in the sample space, while L-R and L-G denote detecting in the latent space using RealNVP and Glow, respectively.

| InD/OOD | S | L-R | L-G |
|---|---|
| FMNIST/MNIST | 1.00 | 1.00 | 1.00 |
| CIFAR-10/SVHN | 0.90 | 0.97 | 0.99 |
| CIFAR-10/CelebA | 0.96 | 0.93 | 0.99 |
| SVHN/CIFAR-10 | 0.62 | 0.98 | 0.99 |
| SVHN/CIFAR-100 | 0.67 | 0.98 | 0.99 |
| SVHN/CelebA | 0.88 | 0.99 | 1.00 |

Table 4: Comparison in the sample and latent spaces with noise. AUROCs are shown and segmented by |. S denotes detection in the sample space, while L-R and L-G denote detecting in the latent space using Real-NVP and Glow, respectively.

| InD/OOD | Noise | S | L-R | L-G |
|---|---|---|
| CIFAR-10/SVHN | No noise | 0.90 | 0.97 | 0.99 |
| | $\mathcal{N}(0, 0.1)$ | 0.86 | 1.00 | 1.00 |
| | $\mathcal{N}(0, 0.2)$ | 0.90 | 1.00 | 1.00 |
| | $\mathcal{N}(0, 0.3)$ | 0.98 | 1.00 | 1.00 |
| | $\mathcal{N}(0, 0.4)$ | 1.00 | 1.00 | 1.00 |
| SVHN/CIFAR-10 | No noise | 0.62 | 0.98 | 0.99 |
| | $\mathcal{N}(0, 0.1)$ | 0.69 | 1.00 | 1.00 |
| | $\mathcal{N}(0, 0.2)$ | 0.79 | 1.00 | 1.00 |
| | $\mathcal{N}(0, 0.3)$ | 0.88 | 1.00 | 1.00 |
| | $\mathcal{N}(0, 0.4)$ | 0.93 | 1.00 | 1.00 |

Table 5: AUPR on RealNVP (higher is better). Our results are denoted by GOD1KS|GOD2KS. Highest AUPR are in boldface, and failure cases (where AUPR is below 0.5) are underlined.

| InD | OOD | batch size = 5 | | | batch size = 10 | | |
|---|---|---|---|---|---|---|---|
| | | TyTest | KLOD | Ours | TyTest | KLOD | Ours |
| FMNIST | MNIST | 0.65 | 0.86 | **0.99**\|**0.99** | 0.68 | 0.89 | **1.00**\|**1.00** |
| | KMNIST | 0.95 | 0.95 | **0.99**\|**0.99** | 0.96 | 0.96 | **1.00**\|**1.00** |
| | Omniglot | 1.00 | 1.00 | 1.00\|1.00 | 1.00 | 1.00 | 1.00\|1.00 |
| CIFAR-10 | SVHN | **0.95** | 0.68 | **0.95**\|0.93 | 0.98 | 0.72 | **1.00**\|0.99 |
| | CelebA | 0.78 | **0.94** | 0.86\|0.86 | 0.87 | **0.99** | 0.94\|0.94 |
| | LSUN | 0.60 | **0.67** | 0.60\|0.61 | 0.70 | **0.78** | 0.66\|0.70 |
| CIFAR-100 | SVHN | **0.96** | 0.63 | **0.96**\|0.94 | 0.99 | 0.68 | **1.00**\|0.99 |
| | CelebA | 0.60 | **0.89** | 0.85\|0.85 | 0.63 | **0.98** | 0.93\|0.93 |
| | LSUN | 0.49 | **0.57** | 0.56\|**0.57** | 0.52 | **0.66** | 0.60\|0.64 |
| SVHN | CIFAR-10 | **1.00** | 0.98 | 0.62\|0.70 | **1.00** | 0.98 | 0.80\|0.89 |
| | CIFAR-100 | **1.00** | 0.98 | 0.64\|0.71 | **1.00** | 0.99 | 0.82\|0.90 |
| | CelebA | **1.00** | **1.00** | 0.79\|0.83 | **1.00** | **1.00** | 0.93\|0.96 |
| | LSUN | **1.00** | 0.99 | 0.70\|0.74 | **1.00** | **1.00** | 0.88\|0.93 |
| CelebA | CIFAR-10 | 0.97 | **0.98** | 0.86\|0.89 | **1.00** | **1.00** | 0.97\|0.98 |
| | CIFAR-100 | **0.98** | **0.98** | 0.85\|0.88 | **1.00** | **1.00** | 0.96\|0.98 |
| | SVHN | 0.83 | 0.85 | **0.98**\|0.97 | 0.86 | 0.99 | **1.00**\|**1.00** |
| | LSUN | **0.99** | **0.99** | 0.81\|0.85 | **1.00** | **1.00** | 0.96\|0.98 |

## G.6 GENERATION QUALITY VS. OOD DETECTION

**Model capacity:** We repeat the experiment regarding generation quality on RealNVP, and the comparison is shown in Figure 8. Again, the batch size is fixed as 10. The observation is similar to Figure 5, i.e. a less capable model that generates more blurred images performs better in OOD detection.

**Training time:** While the OOD detection performance generally improves with training time, as shown in Figure 9, we still found some anomalies. We found a few counterexamples that the OOD detection performance fluctuate or even declines with training time, as shown in Figure 10 and 11.

This counter-intuitive phenomenon might be surprising at a first glance, but it is not completely baseless. Note that the OOD detection failure for FMNIST/MNIST and CIFAR-10/SVHN (the two most famous OOD detection failure cases as reported by Nalisnick et al. (2019a)) is because the generative model gives OOD input higher density estimation than InD, i.e. the log likelihood histogram of OOD sits on the right hand side of InD. A better model (either with a larger capacity

Table 6: AUROC and AUPR on Glow (higher is better). Our results are denoted by GOD1KS|GOD2KS. Highest metrics are denoted in boldface, and failure cases (where metric is below 0.5) are underlined.

| InD | OOD | Metric | batch size = 5 | | | batch size = 10 | | |
|---|---|---|---|---|---|---|---|---|
| | | | TyTest | KLOD | Ours | TyTest | KLOD | Ours |
| FMNIST | MNIST | AUC | **0.99** | 0.12 | 0.19\|0.98 | **1.00** | 0.04 | 0.11\|**1.00** |
| | | AUPR | **0.98** | 0.32 | 0.34\|**0.98** | **1.00** | 0.31 | 0.32\|**1.00** |
| | KMNIST | AUC | 0.51 | 0.38 | 0.01\|**0.97** | 0.55 | 0.13 | 0.00\|**1.00** |
| | | AUPR | 0.53 | 0.41 | 0.31\|**0.96** | 0.56 | 0.33 | 0.30\|**1.00** |
| | Omniglot | AUC | 1.00 | 1.00 | 1.00\|1.00 | 1.00 | 1.00 | 1.00\|1.00 |
| | | AUPR | 1.00 | 1.00 | 1.00\|1.00 | 1.00 | 1.00 | 1.00\|1.00 |
| CIFAR-10 | SVHN | AUC | **1.00** | 0.96 | 0.98\|0.96 | 1.00 | 1.00 | 1.00\|1.00 |
| | | AUPR | **1.00** | 0.98 | 0.99\|0.98 | 1.00 | 1.00 | 1.00\|1.00 |
| | CelebA | AUC | 0.55 | **0.98** | 0.71\|0.86 | 0.68 | **1.00** | 0.79\|0.96 |
| | | AUPR | 0.69 | **0.99** | 0.82\|0.92 | 0.77 | **1.00** | 0.88\|0.98 |
| | LSUN | AUC | **0.86** | 0.82 | 0.31\|0.60 | **0.94** | 0.80 | 0.18\|0.58 |
| | | AUPR | **0.88** | 0.81 | 0.38\|0.58 | **0.95** | 0.78 | 0.33\|0.56 |
| CIFAR-100 | SVHN | AUC | **1.00** | 0.79 | 0.94\|0.90 | **1.00** | 0.97 | 0.99\|0.98 |
| | | AUPR | **1.00** | 0.91 | 0.98\|0.96 | **1.00** | 0.99 | 0.99\|0.99 |
| | CelebA | AUC | 0.35 | **0.97** | 0.74\|0.82 | 0.35 | **1.00** | 0.84\|0.96 |
| | | AUPR | 0.55 | **0.98** | 0.85\|0.90 | 0.55 | **1.00** | 0.91\|0.98 |
| | LSUN | AUC | 0.66 | **0.82** | 0.38\|0.59 | 0.76 | **0.86** | 0.27\|0.59 |
| | | AUPR | 0.66 | **0.80** | 0.42\|0.58 | 0.78 | **0.85** | 0.37\|0.58 |
| SVHN | CIFAR-10 | AUC | **1.00** | **1.00** | 0.10\|0.92 | **1.00** | **1.00** | 0.01\|0.99 |
| | | AUPR | **1.00** | **1.00** | 0.16\|0.84 | **1.00** | **1.00** | 0.15\|0.98 |
| | CIFAR-100 | AUC | **1.00** | **1.00** | 0.08\|0.93 | **1.00** | **1.00** | 0.01\|0.99 |
| | | AUPR | **1.00** | **1.00** | 0.16\|0.86 | **1.00** | **1.00** | 0.15\|0.98 |
| | CelebA | AUC | **1.00** | **1.00** | 0.13\|**1.00** | **1.00** | **1.00** | 0.02\|**1.00** |
| | | AUPR | **1.00** | **1.00** | 0.27\|0.99 | **1.00** | **1.00** | 0.26\|**1.00** |
| | LSUN | AUC | **1.00** | **1.00** | 0.02\|0.99 | **1.00** | **1.00** | 0.00\|**1.00** |
| | | AUPR | **1.00** | **1.00** | 0.15\|0.98 | **1.00** | **1.00** | 0.15\|**1.00** |
| CelebA | CIFAR-10 | AUC | 0.26 | 0.68 | 0.36\|**0.84** | 0.14 | 0.67 | 0.27\|**0.94** |
| | | AUPR | 0.23 | 0.53 | 0.27\|**0.73** | 0.20 | 0.52 | 0.23\|**0.91** |
| | CIFAR-100 | AUC | 0.29 | 0.65 | 0.30\|**0.82** | 0.14 | 0.61 | 0.20\|**0.94** |
| | | AUPR | 0.25 | 0.50 | 0.24\|**0.71** | 0.21 | 0.46 | 0.21\|**0.90** |
| | SVHN | AUC | **1.00** | 0.83 | 0.57\|0.97 | **1.00** | 0.88 | 0.53\|**1.00** |
| | | AUPR | **1.00** | 0.87 | 0.67\|0.98 | **1.00** | 0.91 | 0.65\|**1.00** |
| | LSUN | AUC | 0.29 | 0.64 | 0.33\|**0.85** | 0.18 | 0.59 | 0.22\|**0.96** |
| | | AUPR | 0.25 | 0.47 | 0.25\|**0.75** | 0.21 | 0.43 | 0.22\|**0.92** |

Table 7: Comparison of KS test and CvM test on RealNVP. AUROC is shown (higher is better). Our results are denoted by GOD1KS|GOD2KS.

| InD/OOD | batch size = 5 | | batch size = 10 | | batch size = 20 | |
|---|---|---|---|---|---|---|
| | KS | CvM | KS | CvM | KS | CvM |
| CIFAR-10/SVHN | 0.86\|0.82 | 0.77\|0.71 | 0.99\|0.97 | 0.97\|0.94 | 1.00\|1.00 | 1.00\|1.00 |
| CIFAR-10/CelebA | 0.80\|0.80 | 0.82\|0.82 | 0.92\|0.93 | 0.94\|0.95 | 0.99\|0.99 | 1.00\|1.00 |
| CIFAR-10/LSUN | 0.61\|0.63 | 0.63\|0.64 | 0.65\|0.68 | 0.68\|0.70 | 0.72\|0.78 | 0.76\|0.79 |
| SVHN/CIFAR-10 | 0.89\|0.93 | 0.88\|0.90 | 0.95\|0.98 | 0.93\|0.96 | 0.99\|1.00 | 0.97\|0.99 |

or more training time) tends to give a more accurate estimation of density. However, the OOD detection is mainly based on both the separation of the InD score and OOD score and their relative order (i.e. InD must have higher likelihood estimation than OOD input, otherwise detection based on the likelihood will fail) instead of their respective accuracy. Some prior works (Schirrmeister et al., 2020; Zhang et al., 2021) reported a related result. Specifically, Schirrmeister et al. (2020) modifies the model architecture, so that the wrong relative order of InD and OOD density histograms is reversed, which leads to the success of OOD detection, although the modified model has worse log likelihood. In contrast, Zhang et al. (2021) consider a successful case, where the relative order of InD

Table 8: AUROC and AUPR when batch size = 20. Our results are denoted by GOD1KS|GOD2KS. We use the boldface for the highest metrics, and underline the detection failure case (where metric is below 0.5).

| InD | OOD | Metric | RealNVP, batch size = 20 | | | Glow, batch size = 20 | | |
|---|---|---|---|---|---|---|---|---|
| | | | TyTest | KLOD | Ours | TyTest | KLOD | Ours |
| FMNIST | MNIST | AUC | 0.83 | 0.97 | **1.00\|1.00** | **1.00** | 0.01 | 0.05\|**1.00** |
| | | AUPR | 0.67 | 0.93 | **1.00\|1.00** | 0.99 | 0.31 | 0.31\|**1.00** |
| | KMNIST | AUC | 0.98 | 0.99 | **1.00\|1.00** | 0.58 | 0.02 | 0.00\|**1.00** |
| | | AUPR | 0.97 | 0.98 | **1.00\|1.00** | 0.60 | 0.31 | 0.31\|**1.00** |
| | Omniglot | AUC | 1.00 | 1.00 | 1.00\|1.00 | 1.00 | 1.00 | 1.00\|1.00 |
| | | AUPR | 1.00 | 1.00 | 1.00\|1.00 | 1.00 | 1.00 | 1.00\|1.00 |
| CIFAR-10 | SVHN | AUC | 0.99 | 0.46 | **1.00\|1.00** | 1.00 | 1.00 | 1.00\|1.00 |
| | | AUPR | **1.00** | 0.78 | **1.00\|1.00** | 1.00 | 1.00 | 1.00\|1.00 |
| | CelebA | AUC | 0.91 | **1.00** | 0.98\|0.99 | 0.78 | **1.00** | 0.89\|**1.00** |
| | | AUPR | 0.94 | **1.00** | 0.99\|0.99 | 0.85 | **1.00** | 0.93\|**1.00** |
| | LSUN | AUC | 0.81 | **0.91** | 0.73\|0.79 | **0.99** | 0.75 | 0.07\|0.55 |
| | | AUPR | 0.80 | **0.90** | 0.75\|0.80 | **0.99** | 0.72 | 0.33\|0.54 |
| CIFAR-100 | SVHN | AUC | **1.00** | 0.41 | **1.00\|1.00** | 1.00 | 1.00 | 1.00\|1.00 |
| | | AUPR | **1.00** | 0.75 | **1.00\|1.00** | 1.00 | 1.00 | 1.00\|1.00 |
| | CelebA | AUC | 0.53 | **1.00** | 0.98\|0.98 | 0.32 | **1.00** | 0.94\|**1.00** |
| | | AUPR | 0.66 | **1.00** | 0.99\|0.99 | 0.54 | **1.00** | 0.97\|**1.00** |
| | LSUN | AUC | 0.60 | **0.85** | 0.69\|0.76 | 0.87 | **0.91** | 0.17\|0.63 |
| | | AUPR | 0.58 | **0.81** | 0.69\|0.76 | 0.88 | **0.91** | 0.34\|0.62 |
| SVHN | CIFAR-10 | AUC | **1.00** | 0.99 | 0.99\|**1.00** | **1.00** | **1.00** | 0.00\|**1.00** |
| | | AUPR | **1.00** | 0.98 | 0.94\|0.99 | **1.00** | **1.00** | 0.15\|**1.00** |
| | CIFAR-100 | AUC | **1.00** | **1.00** | 0.99\|**1.00** | **1.00** | **1.00** | 0.00\|**1.00** |
| | | AUPR | **1.00** | 0.99 | 0.95\|0.99 | **1.00** | **1.00** | 0.15\|**1.00** |
| | CelebA | AUC | 1.00 | 1.00 | 1.00\|1.00 | **1.00** | **1.00** | 0.00\|**1.00** |
| | | AUPR | 1.00 | 1.00 | 1.00\|1.00 | **1.00** | **1.00** | 0.26\|**1.00** |
| | LSUN | AUC | 1.00 | 1.00 | 1.00\|1.00 | **1.00** | **1.00** | 0.00\|**1.00** |
| | | AUPR | **1.00** | **1.00** | 0.99\|**1.00** | **1.00** | **1.00** | 0.16\|**1.00** |
| CelebA | CIFAR-10 | AUC | 1.00 | 1.00 | 1.00\|1.00 | 0.05 | 0.67 | 0.17\|**1.00** |
| | | AUPR | 1.00 | 1.00 | 1.00\|1.00 | 0.19 | 0.54 | 0.21\|**0.99** |
| | CIFAR-100 | AUC | 1.00 | 1.00 | 1.00\|1.00 | 0.05 | 0.57 | 0.10\|**0.99** |
| | | AUPR | 1.00 | 1.00 | 1.00\|1.00 | 0.19 | 0.42 | 0.20\|**0.99** |
| | SVHN | AUC | 0.85 | **1.00** | **1.00\|1.00** | **1.00** | 0.93 | 0.50\|**1.00** |
| | | AUPR | 0.89 | **1.00** | **1.00\|1.00** | **1.00** | 0.95 | 0.62\|**1.00** |
| | LSUN | AUC | 1.00 | 1.00 | 1.00\|1.00 | 0.10 | 0.58 | 0.12\|**1.00** |
| | | AUPR | 1.00 | 1.00 | 1.00\|1.00 | 0.20 | 0.44 | 0.20\|**1.00** |

and OOD bpd (bits per dimension) histograms is correct, but they found that a mis-estimated model can lead to a smaller overlap in bpd histograms of InD and OOD, which results in the improvement of detection performance (see Figure 2 in Zhang et al. (2021)). This intuition is also valid in our method. Take our one-sample test as an example. For a worse model, the transformed distribution of both InD and OOD in the latent space will be further away from the prior, compared to a better model, but the separation in this case might be more pronounced, which is why a worse model can sometimes lead to better OOD detection.

More importantly, our detection method is not based on likelihood score, and yet we still observe a similar phenomenon. These results are indeed related, and our results corroborate existing work by examining more flow models, test statistics, real datasets and metrics.

## G.7 INTRA-DATASET OOD DETECTION

Till now, all our OOD detection experiments were conducted across different datasets (i.e. inter-dataset OOD). We conduct additional experiments on detecting OOD within one dataset, i.e. treating one class as InD, the rest classes combined as OOD. Table 9 shows the detection results on FMNIST and CIFAR-10. The batch size is fixed as 10, and our method uses 200 random projections. Compared to SOTA baselines, our method achieves the best intra-dataset OOD detection performance on FMNIST dataset for all classes, while on CIFAR-10 dataset, our method yields the near-best de-

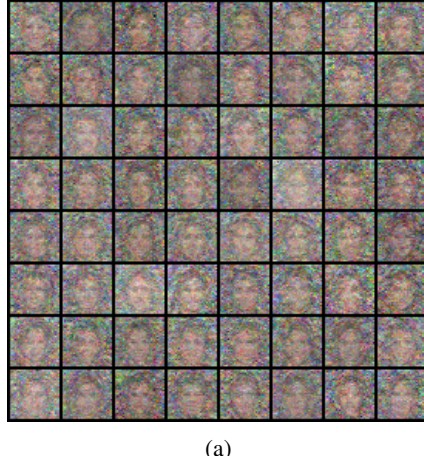 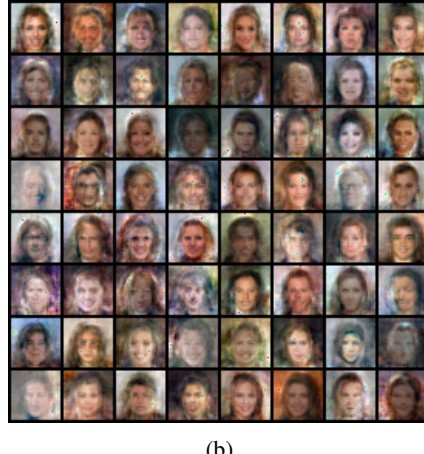

|                          |                          |
| :----------------------: | :----------------------: |
| (a)                      | (b)                      |

Figure 8: Model capacity vs. OOD detection. Images are generated from RealNVP trained on CelebA. Assume CIFAR-10 as OOD, and our GOD1KS and GOD2KS results are segmented by |. (a) Generated images from a RealNVP with 1 block and 2048 hidden channels. AUROCs = 0.98|0.99. (b) Generated images from a RealNVP with 16 blocks and 512 hidden channels. AUROCs = 0.88|0.88.

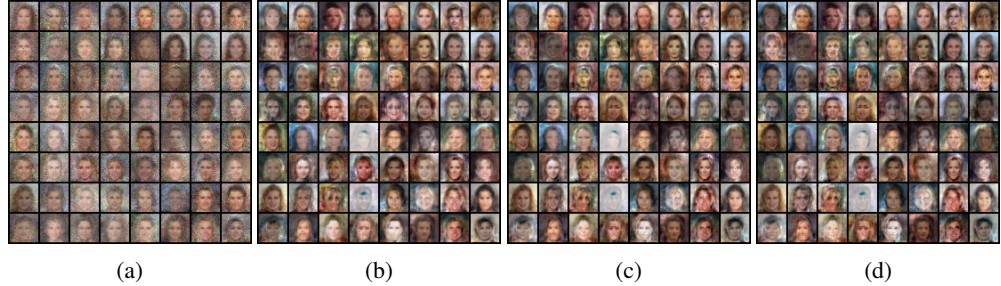

|     (a)     |     (b)     |     (c)     |     (d)     |
| :---------: | :---------: | :---------: | :---------: |

Figure 9: Training epochs vs. OOD detection. Images are generated from RealNVP trained on CelebA. Assume CIFAR-10 as OOD, and our GOD1KS and GOD2KS results are segmented by |. (a) Trained for 10 epochs. AUROCs = 0.78|0.78. (b) Trained for 100 epochs. AUROCs = 0.81|0.81. (c) Trained for 150 epochs. AUROCs = 0.86|0.86. (d) Trained for 200 epochs. AUROCs = 0.88|0.88.

tection performance for all classes. Besides, there is no detection failure case in our method, which serves as another evidence of the effectiveness and generality of our proposed method.

### G.8  OOD DETECTION ON TIME-SERIES DATA

Till now, all our experiments are conducted on widely used image benchmarks. It will be interesting to investigate whether our method can generalize beyond image datasets. We consider testing physiological signals (time-series data), e.g. Electrocardiography (ECG) and Electroencephalography (EEG). Clinically, ECG and EEG are electric signals that could reflect heart or brain activities, thus can be critical for the monitoring and diagnosis of cardiac dysfunctions or neural disorders. In this experiment, we train a RealNVP on one signal, and test the OOD detection performance by treating the other kind of signal as OOD. Our experimental data come from a publicly available database, i.e. MIT-BIH Polysomnographic database (Ichimaru & Moody, 1999; Goldberger et al., 2000) [3]. The MIT-BIH Polysomnographic Database collects various physiological signals, including EEG and ECG, from 16 subjects during sleep. We randomly select four records from the database for analysis, i.e. slp03, slp04, slp14, slp16, where each record is exactly 6-hour long. Amplitudes are normalized between 0 and 1. The whole signal is segmented into 10-second strips, so the size of

---

[3]https://physionet.org/content/slpdb/1.0.0/

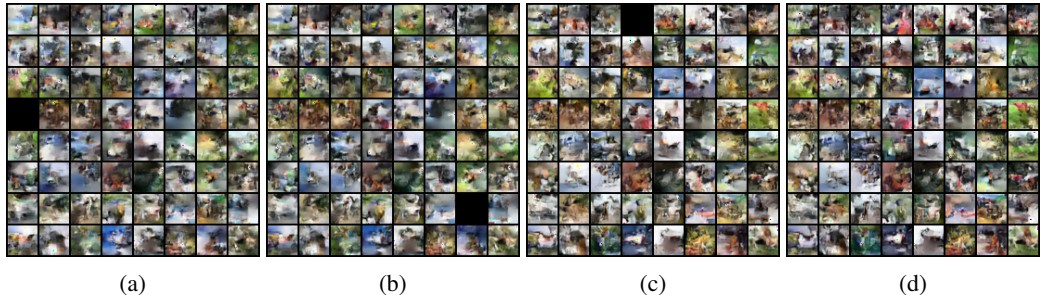

|  |  |  |  |
|:---:|:---:|:---:|:---:|
| (a) | (b) | (c) | (d) |

Figure 10: Training epochs vs. OOD detection. Images are generated from a Glow with $K = 16, L = 3, h = 128$ trained on CIFAR-10. Treating SVHN as OOD, our GOD1KS and GOD2KS results are segmented by |. (a) Trained for 5 epochs. AUROCs = 0.96|0.96. (b) Trained for 10 epochs. AUROCs = 0.96|0.95. (c) Trained for 100 epochs. AUROCs = 0.76|0.84. (d) Trained for 200 epochs. AUROCs = 0.87|0.84.

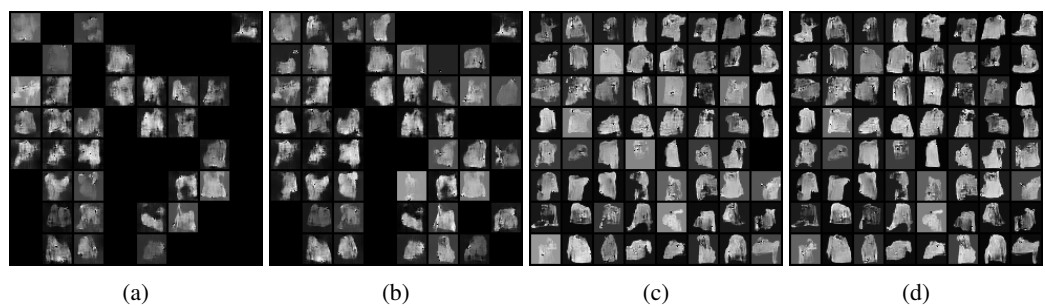

|  |  |  |  |
|:---:|:---:|:---:|:---:|
| (a) | (b) | (c) | (d) |

Figure 11: Training epochs vs. OOD detection. Images are generated from a Glow with $K = 16, L = 2, h = 128$ trained on FMNIST. Treating MNIST as OOD, our GOD1KS and GOD2KS results are segmented by |. (a) Trained for 5 epochs. AUROCs = 1.00|1.00. (b) Trained for 10 epochs. AUROCs = 1.00|1.00. (c) Trained for 100 epochs. AUROCs = 0.90|1.00. (d) Trained for 200 epochs. AUROCs = 0.92|1.00.

this toy dataset is 8640, where 10% is randomly held out for validation, and another 10% is randomly held out for test. The rest 80% is used for training. The sampling rate is 250 Hz, so the data dimension is 2500. Table 10 shows the detection performance of OOD detection on ECG/EEG and EEG/ECG by our method as well as two SOTA baselines, where the batch size is fixed as 10 and we use 200 random projections for our methods. It can be observed from Table 10 that our method, especially GOD2KS, indicates more robustness against SOTA baselines. In the EEG/ECG setting, typicality test fails, implying an overlap in density histograms of InD and OOD, and KLOD is slightly better than random guess. In contrast, our GOD2KS achieves around 0.9 metrics in both experimental settings.

## G.9 INVESTIGATION OF CIFARS VS. LSUN

For CIFAR vs LSUN, our experimental results indicate that this is a challenging case for our method. We examined the raw distributions of the two datasets, through per-dimensional mean and variance. The detailed result is given in Figure 12, where we verify that CIFARs and LSUN share a very close per-dimensional mean and variance, while for CIFARs vs. SVHN, although their per-dimensional mean share a similar range, their distributional shapes are different in terms of per-dimensional variances. This implies the CIFARs vs. LSUN pair is distributionally more similar and hence more challenging for our methods. The immediate solution is to increase the batch size. When batch size = 100, our methods (using RealNVP) can achieve AUROC of 0.97|1.00 for CIFAR-10 vs. LSUN, and 0.90|0.99 for CIFAR-100 vs. LSUN. When batch size is beyond 200, our methods can achieve perfect detection performance in both settings.

Table 9: Intra-dataset OOD detection using RealNVP. AUROC and AUPR are shown (higher is better). The batch size is fixed as 10. Our results are denoted by GOD1KS|GOD2KS.

| Classes | Metrics | FMNIST | | | CIFAR-10 | | |
|---|---|---|---|---|---|---|---|
| | | TyTest | KLOD | Ours | TyTest | KLOD | Ours |
| 0 | AUC | 0.99 | **1.00** | **1.00\|1.00** | 0.93 | **0.98** | 0.94\|0.97 |
| | AUPR | 1.00 | 1.00 | 1.00\|1.00 | 0.99 | **1.00** | 0.99\|0.99 |
| 1 | AUC | 1.00 | 1.00 | 1.00\|1.00 | 0.50 | 0.60 | **0.69**\|0.68 |
| | AUPR | 1.00 | 1.00 | 1.00\|1.00 | 0.89 | 0.93 | **0.95**\|0.94 |
| 2 | AUC | 0.96 | 0.97 | **0.99\|0.99** | **0.88** | 0.83 | 0.75\|0.79 |
| | AUPR | 0.99 | **1.00** | **1.00\|1.00** | **0.98** | 0.97 | 0.95\|0.96 |
| 3 | AUC | 1.00 | 1.00 | 1.00\|1.00 | 0.61 | **0.68** | 0.63\|0.63 |
| | AUPR | 1.00 | 1.00 | 1.00\|1.00 | 0.93 | **0.94** | 0.93\|0.93 |
| 4 | AUC | 0.98 | 0.98 | **1.00\|1.00** | **0.98** | 0.97 | 0.93\|0.94 |
| | AUPR | 1.00 | 1.00 | 1.00\|1.00 | **1.00** | **1.00** | 0.99\|0.99 |
| 5 | AUC | 1.00 | 1.00 | 1.00\|1.00 | 0.68 | **0.80** | 0.75\|0.75 |
| | AUPR | 1.00 | 1.00 | 1.00\|1.00 | 0.95 | **0.97** | 0.96\|0.96 |
| 6 | AUC | 0.88 | 0.90 | 0.95\|**0.96** | 0.89 | **0.93** | 0.91\|0.92 |
| | AUPR | 0.98 | 0.99 | 0.99\|**1.00** | 0.99 | 0.99 | 0.99\|0.99 |
| 7 | AUC | 1.00 | 1.00 | 1.00\|1.00 | 0.58 | 0.64 | **0.66**\|0.65 |
| | AUPR | 1.00 | 1.00 | 1.00\|1.00 | 0.93 | **0.94** | **0.94\|0.94** |
| 8 | AUC | 0.74 | 0.86 | **0.99\|0.99** | **0.99** | 0.97 | 0.96\|0.97 |
| | AUPR | 0.94 | 0.98 | **1.00\|1.00** | 1.00 | 1.00 | 1.00\|1.00 |
| 9 | AUC | 1.00 | 1.00 | 1.00\|1.00 | 0.51 | 0.81 | **0.82**\|0.81 |
| | AUPR | 1.00 | 1.00 | 1.00\|1.00 | 0.90 | **0.98** | **0.98\|0.98** |

Table 10: OOD detection on EEG and ECG using RealNVP. AUROC and AUPR are shown (higher is better). The batch size is fixed as 10. Our results are denoted by GOD1KS|GOD2KS.

| InD/OOD | TyTest | | KLOD | | Ours | |
|---|---|---|---|---|---|---|
| | AUC | AUPR | AUC | AUPR | AUC | AUPR |
| EEG/ECG | 0.15 | 0.33 | 0.60 | 0.50 | 0.50\|0.92 | 0.47\|0.88 |
| ECG/EEG | 1.00 | 1.00 | 0.99 | 0.99 | 0.90\|0.92 | 0.92\|0.94 |

## G.10 COMPARISON WITH AVERAGE DENSITY

For thoroughness we also include comparison against a straightforward baseline, i.e. using the averaged density across a batch as the detection score (we denote this approach as Avg in Table 11). We conduct additional experiments on FMNIST/MNIST, CIFAR-10/SVHN (the two most popular experimental settings in OOD detection) using both RealNVP and Glow, as shown in Table 11. The batch size is fixed as 10. The results of other methods are directly cited from Table 1 and Table 6. It can be observed that directly averaging the density across a batch is not competitive. The result is not surprising, because as long as the density histograms of InD and OOD sit in the wrong relative position, the average density will not work, regardless of batch size.

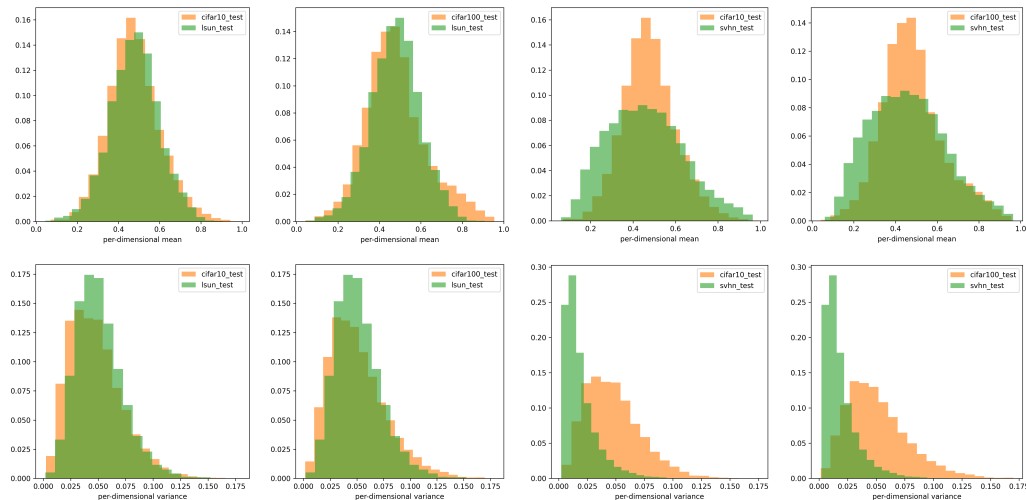

(a) CIFAR-10 vs. LSUN   (b) CIFAR-100 vs. LSUN   (c) CIFAR-10 vs. SVHN   (d) CIFAR-100 vs. SVHN

Figure 12: Per-dimensional mean and variance comparison. (a)-(b) Per-dimensional mean and variance of CIFARs and LSUN are close, which implies that CIFARs and LSUN share a similar distribution of normalized pixel values. (c)-(d) As a comparison, CIFARs vs SVHN have more clearly separable per-dimensional variance.

Table 11: Comparison with averaged density. Our results are denoted by GOD1KS|GOD2KS. We use the boldface for the highest metric, and underline the detection failure case (where metric is below 0.5).

| InD/OOD | Metrics | RealNVP, batch size = 10 | | | | Glow, batch size = 10 | | | |
|---|---|---|---|---|---|---|---|---|---|
| | | Avg | TyTest | KLOD | Ours | Avg | TyTest | KLOD | Ours |
| FMNIST/MNIST | AUC | 0.83 | 0.82 | 0.96 | **1.00\|1.00** | 0.00 | **1.00** | 0.04 | 0.11\|**1.00** |
| | AUPR | 0.69 | 0.68 | 0.89 | **1.00\|1.00** | 0.31 | **1.00** | 0.31 | 0.32\|**1.00** |
| CIFAR-10/SVHN | AUC | 0.01 | 0.95 | 0.35 | **0.99**\|0.98 | 0.00 | **1.00** | **1.00** | **1.00\|1.00** |
| | AUPR | 0.51 | 0.98 | 0.72 | **1.00**\|0.99 | 0.51 | **1.00** | **1.00** | **1.00\|1.00** |

