# OpenReview forum: "Revisiting flow generative models for Out-of-distribution detection"
_ICLR.cc/2022/Conference — ICLR 2022 Poster_

### Official Review · Reviewer_XXZp · 2021-10-30

**Correctness:** 4
**Technical Novelty And Significance:** 2
**Empirical Novelty And Significance:** 3
**Recommendation:** 8
**Confidence:** 4

**Main Review:**

I found it easy to understand the ideas and methods of the paper, results are also well-understandable.

Overall, the paper does not bring too many surprises from my view - I would have expected any use of 1d projections for OOD detection to work better in latent space than in input space. Also it seems not too surprising that this very generic method cannot result in very high OOD detection performance (compared to methods exploiting more assumptions about OOD data). But in any case good to see precisely how well OOD detection works in this way. The main surprising part might be that generation quality can be negatively correlated with OOD detection performance.

The manuscript would benefit from discussing other OOD detection approaches in latent space, https://arxiv.org/abs/2006.10848 also present one method using parts of the latent space of the Glow model, https://arxiv.org/abs/2102.08248 uses VAEs, not flows, but also look at parts of the latent space there. In the future, could be interesting to compare to such methods (using parts of latent space/log-likelihood contributions), which exploit more assumptions about where semantic differences are encoded in the latent space, on both semantic and non-semantic OOD detection. Note also that https://arxiv.org/abs/2006.10848 report a related result to the result that generation quality and OOD detection performance can be negatively correlated, finding that fully connected flow models have worse log likelihoods but better OOD detection performance (see Sec 2).

I would like it if caption figures were more self-contained and figures could be understood separately from the text. E.g., figure 3 and figure 5 captions could also talk a bit more about what one could learn from the figures.

**Summary Of The Paper:**

The paper proposes evaluates using statistical tests on random 1d-projections in the latent space of a flow model for groupwise out-of-distribution (OOD) detection. Concretely, Komolgorov-Smirnov tests are used to compare latent encodings of a given batch of samples to the expected distribution of the in-distribution latent encodings. For the expected distribution, either the predefined latent prior is used, resulting in a 1-sample Komolgorov-Smirnov test, or the empirical distribution of the latent encodings of the training data  is used, resulting in a 2-sample Komolgorov-Smirnov test. The paper reports similar groupwise OOD detection performance as existing methods, with the 2-sample KS-test method having less failure cases.

**Summary Of The Review:**

The paper in my view does not present too many surprising results, but I also find it valuable to see non-surprising results that are clearly presented and easy to understand. And the result on generation quality and OOD detection performance being negatively correlated is also valuable to know.

**Update Post Rebuttal**
The revisions have further improved the manuscript, I increase my score to 8.

---

> ### Author Response · Authors · 2021-11-23
> **Response to Reviewer XXZp**
>
> Thank you very much for your recognition of our contributions, and we address your questions/concerns below:
>
> **Q1.** The manuscript would benefit from discussing other OOD detection approaches in latent space, [1] also present one method using parts of the latent space of the Glow model, [2] uses VAEs, not flows, but also look at parts of the latent space there.
>
> **Response:** Thank you for bringing these important references to our attention. We have cited both works and added the following discussion to the revision:
> > Note that [1,2] also develop detection scores based on (partial) latent variables of flow model and VAE, respectively. Differently, [2] interpret their likelihood ratio test as comparing divergence between true and approximate posterior in latent space (since true posterior is intractable), whereas [1] contrast the log likelihood of the last-scale latent code that encodes global object-specific features with that of some background latent code. Nevertheless, both approaches do not employ random projections or KS tests as we do.
>
> **Q2.** Note also that [1] report a related result to the result that generation quality and OOD detection performance can be negatively correlated, finding that fully connected flow models have worse log likelihoods but better OOD detection performance (see Sec 2).
>
> **Response:** Thank you for this comment. We think [1], as well as [3], achieved OOD detection performance improvement by the change in log likelihood. Note that the OOD detection failure for FMNIST/MNIST and CIFAR-10/SVHN, as reported by [4], is because the generative model gives OOD inputs higher likelihood estimate than InD ones, i.e. the log likelihood histogram of OOD sits on the right hand side of InD. [1] modified the model architecture, so that the OOD likelihood histogram sits on the left hand side of InD, which leads to the success of OOD detection, although the modified model has worse log likelihood. In contrast, [3] considered a successful case, where the relative order of InD and OOD bpd (bits per dimension) histograms is correct, but they found that a mis-estimated model can lead to a smaller overlap in InD and OOD bpd histograms, which results in the improvement of detection performance. Our results are different, where neither the overlap nor relative position of histograms is significantly changed (see our Figure 5 for example). More importantly, our detection method is not based on likelihood score, and yet we still observe a similar phenomenon. These results are indeed related, and our results corroborate existing work by examining more flow models, test statistics, real datasets and metrics.
>
> **Q3.** I would like it if caption figures were more self-contained and figures could be understood separately from the text. E.g., figure 3 and figure 5 captions could also talk a bit more about what one could learn from the figures.
>
> **Response:** Thank you for this suggestion. We've added more details to our figure captions on the revision.
>
> **References**
>
> [1] Robin Schirrmeister, Yuxuan Zhou, Tonio Ball, and Dan Zhang. Understanding anomaly detection with deep invertible networks through hierarchies of distributions and features. In Advances in Neural Information Processing Systems, volume 33, 2020.
>
> [2] Jakob D. Drachmann Havtorn, Jes Frellsen, Soren Hauberg, and Lars Maaløe. Hierarchical vaes know what they don’t know.  In Proceedings of the 38th International Conference on Machine Learning, volume 139, pp. 4117–4128,2021.
>
> [3] Lily Zhang, Mark Goldstein, and Rajesh Ranganath. Understanding failures in out-of-distribution detection with deep generative models. In International Conference on Machine Learning, pp. 12427–12436, 2021.
>
> [4] Eric Nalisnick, Akihiro Matsukawa, Yee Whye Teh, Dilan Gorur, and Balaji Lakshminarayanan.  Do deep generative models know what they don’t know?  In International Conference on Learning Representations, 2019.

---

> > ### Comment · Reviewer_XXZp · 2021-11-30
> > **Thanks; increased score**
> >
> > Thanks for your revisions, in my view they have improved the manuscript, therefore I increased the score to accept.

---

### Official Review · Reviewer_R73v · 2021-11-02

**Correctness:** 4
**Technical Novelty And Significance:** 3
**Empirical Novelty And Significance:** 3
**Recommendation:** 6
**Confidence:** 4

**Main Review:**

Strengths:

S1. The proposed approach is interesting, simple, mathematically sound and effective. In comparison with Zhang et al. (2020), the proposed approach does not require estimation of a multivariate distribution.

S2. Experiments show robust overall performance. Notably, there are less failure cases than in related methods.

Weaknesses:

W1: Authors does not propose a compelling motivation for groupwise OOD detection.

W2: The evaluation should also show the OOD AP metric as in Zhang et al. (2020).

Suggestions
- Can you offer an intuitive explanation for the failures of TyTest and KLOD in Table 1?
- It may be interesting to evaluate the average density across the input batch as a straight-forward baseline. I do not expect this to be competitive, but averaging may improve over the pointwise result.
- Can you provide any insight why the method can not exploit normalizing flows with large capacity?
- Include algorithmic presentation of GOD2KS in the supplement (for completeness).
- Page 2: absolute value of the Jacobian -> absolute value of the determinant of the Jacobian.
- Figure 1, caption: explain S_0 and S_1.
- Page 3, "in order to evade the curse of dimensionality": this becomes especially relevant in contemporary normalizing flows (eg. VFlow, DenseFlow) where latent dimensionality is greater than input dimensionality.
- Page 3, "is distinct from the distributions": make clear that this refers only to Fig1(b).
- Figure 2, caption: does not follow -> follow.


**Summary Of The Paper:**

The manuscript addresses groupwise outlier detection with normalizing flows. The main idea is to project latent representations of the input batch onto several random directions and to evaluate the resulting 1D distributions according to the Kolmogorov-Smirnov test either with respect to the prior (GOD1KS) or with respect to the corresponding distribution of the training data (GOD2KS). Finally, OOD ranking is performed according to the average KS value across all random directions. GOD1KS is used when the underlying normalizing flow is well learned, while GOD2KS is an option when there is some uncertainty regarding the learning outcome. Experimental performance decreases  when the method is used in combination with undertrained normalizing flow or a normalizing flow with a larger capacity.

**Summary Of The Review:**

This paper provides interesting insights about normalizing flows and OOD detection.

---

> ### Author Response · Authors · 2021-11-23
> **Response to Reviewer R73v - continued**
>
> **Q6.** Include algorithmic presentation of GOD2KS in the supplement (for completeness).
>
> **Response:** Thank you for this suggestion. We have added it (Algorithm 2) in Appendix B.
>
> **Q7.** Page 2: absolute value of the Jacobian $\rightarrow$ absolute value of the determinant of the Jacobian.
>
> **Response:** We have fixed it. Thank you.
>
> **Q8.** Figure 1, caption: explain $S_0$ and $S_1$.
>
> **Response:** Thank you for pointing this out. $S_0$ and $S_1$ represent the two latent dimensions in this example. We have changed them to $z_0$ and $z_1$ for clarity.
>
> **Q9.** Page 3, "in order to evade the curse of dimensionality": this becomes especially relevant in contemporary normalizing flows (eg. VFlow, DenseFlow) where latent dimensionality is greater than input dimensionality.
>
> **Response:** Thank you for this excellent comment! Indeed, when latent dimension is even larger than input dimension, we need to more carefully deal with the curse of dimensionality. We include the references as you suggest in page 3.
>
> **Q10.** Page 3, ``is distinct from the distributions": make clear that this refers only to Fig1(b).
>
> **Response:**  We have made this clear. Thank you for pointing it out.
>
> **Q11.** Figure 2, caption: does not follow $\rightarrow$ follow
>
> **Response:** We have fixed it. Thank you.

---

> ### Author Response · Authors · 2021-11-23
> **Response to Reviewer R73v**
>
> Thank you very much for your detailed comments, and we address your questions/concerns below:
>
> **Q1.** Authors does not propose a compelling motivation for groupwise OOD detection.
>
> **Response:** Thank you for this question. We summarize our motivation for the groupwise OOD detection as follows. According to [1], group anomaly can be characterized as (1) point based, where each individual point in the group is anomalous, and (2) distribution based, where a group of points differs from the regular pattern while any single point in the group may seem regular. Therefore, distribution based group anomaly is only detectable by GOD methods. Practical applications include detecting Higgs Bosons as a group of collision events in high-energy particle physics [2], and detecting distributed denial-of-services attacks via a group of multi-sensor networks [3].
> In the revision, we've included the above explanation in Section 2.1. Also, we've constructed a synthetic Gaussian experiment to illustrate the above point (see our Appendix.E for details).
>
> **Q2.** The evaluation should also show the OOD AP metric as in Zhang et al. (2020).
>
> **Response:** Thank you very much for this suggestion! In the revision, we've added the AUPR metric in our results on all datasets (See Tables 5, 6, 8). With the additional AUPR metric our main claim remains unchanged, i.e., GOD2KS is more robust and generally satisfactory across all listed benchmark datasets.
>
> **Q3.** Can you offer an intuitive explanation for the failures of TyTest and KLOD in Table 1?
>
> **Response:** Thank you for the suggestion! Our intuition for the failures of TyTest and KLOD in Table 1 is as follows. For TyTest, when the log-density histograms of InD and OOD overlap TyTest may not work well, which has also been mentioned in the original TyTest paper. For KLOD, as we compute $\mu^\top\mu$ as the score, when OOD has lower latent values $\mu$ than InD, then KLOD may fail.
>
> **Q4.** It may be interesting to evaluate the average density across the input batch as a straight-forward baseline. I do not expect this to be competitive, but averaging may improve over the pointwise result.
>
> **Response:**  In the revision, we've included experiments on FMNIST/MNIST and CIFAR-10/SVHN (the two most popular OOD detection cases) for both RealNVP and Glow using the metric the reviewer suggested. The result is shown in Table 11 in the new Appendix G.9. We confirm that the average density is indeed not a competitive baseline. This is not surprising, because as long as the density histograms of both InD and OOD sit in the wrong relative position, the average density will not work, regardless of the batch size.
>
> **Q5.** Can you provide any insight why the method can not exploit normalizing flows with large capacity?
>
> **Response:** Thank you for this suggestion! A better model (either with a larger capacity or more training time) tends to give a more accurate estimation of density. However, the OOD detection is mainly based on both the separation of the InD score and OOD score and their relative order (e.g. when using likelihood as detection score, InD must have higher likelihood than OOD input, otherwise the detection based on likelihood will fail) instead of their respective accuracy. Figure 2 in [4] shows such an example, where the mis-estimated model leads to a worse log-likelihood estimation (the relative order of InD and OOD likelihood is correct), but it results in a larger separation between the density histograms of InD and OOD. This intuition is also valid in our method. Take our one-sample test as an example. For a worse model, the transformed distribution of both InD and OOD in the latent space will be further away from the prior, compared to a better model, but the separation in this case might be more pronounced, which is why a worse model can sometimes lead to  better OOD detection.
>
>
> **References**
>
> [1] Liang Xiong, Barnabas Poczos, Jeff Schneider, Andrew Connolly, and Jake VanderPlas.   Hierarchical probabilistic models  for  group  anomaly  detection.   In International  Conference on  Artificial  Intelligence  and  Statistics,  vol-ume 15, pp. 789–797, 2011.
>
> [2] K Muandet and B Sch ̈olkopf.  One-class support measure machines for group anomaly detection.  In 29th Conference on Uncertainty in Artificial Intelligence, pp. 449–458, 2013.
>
> [3] Xiaofan Chen and Shunzheng Yu. A collaborative intrusion detection system against ddos for sdn. IEICE Transactionson Information and Systems, 99(9):2395–2399, 2016.
>
> [4] Lily Zhang, Mark Goldstein, and Rajesh Ranganath. Understanding failures in out-of-distribution detection with deep generative models. In International Conference on Machine Learning, pp. 12427–12436, 2021.

---

> > ### Comment · Reviewer_R73v · 2021-11-26
> > **Motivation, title**
> >
> > Thank you on interesting discussion!
> >
> > The proposed motivation is somewhat misaligned with the presented experiments on natural images. Still, new experiments on EEG/ECG signals confirm applicability of the proposed method for real-world problems.
> >
> > It would be a good idea to correct the title to avoid disappointment of the readers interested in point-wise OOD with normalizing flows. A more informative title would be simply: REVISITING FLOW GENERATIVE MODELS FOR GROUP-WISE OUT-OF-DISTRIBUTION DETECTION
> >
> > I am leaning towards keeping my original rating (weak accept).

---

> > > ### Author Response · Authors · 2021-11-26
> > > **Thanks!**
> > >
> > > Thank you for your suggestion! We agree, and we have changed the title as you suggest, although the change cannot be updated on openreview yet.

---

### Official Review · Reviewer_Kij3 · 2021-11-04

**Correctness:** 3
**Technical Novelty And Significance:** 3
**Empirical Novelty And Significance:** 2
**Recommendation:** 8
**Confidence:** 3

**Main Review:**

Strengths

The proposed approach is simple and does not require training additional components. It can be applied both in data space and latent of the flow, but according to experimental studies application of latent space gives better results. The results are comparable to the results obtained to the reference approaches - the significant gain is observed on CIFAR10 vs SVHN setting. The paper is easy to follow and the idea of the paper is clear. The proposed approach is easy to applied so the results are easy to reproduce.

Weaknesses

The contribution of this work is smart but novelty is a bit limited due to the fact that the authors combine two known approaches - KS statistical test and random projections. It seems that this framework is general and can be even applied to the bottleneck models like VAE or even for any low dimensional features extracted from the data. The question is can we apply this framework to any low dimensional representation of data or we need the invertible mapping that does not decrease the dimension and direct access to the likelihood.

The second question is about Carleman’s condition. Considering the case where we use flow’s latent is it sufficient that it would be satisfied by Gaussian (flow latent distribution) or it should be satisfied by data distribution (the normalised Gaussian by determinant of the Jacobian)?

The third issue is about the results. The model seems to be comparable to KLOD besides the CIFAR10 vs SVHN where it performs better. On the other hand, it has problems with CIFAR vs LeSUN setting. Do you have some intuition what is the reason and how to deal with that issue? Most of the experiments that examine the OOD problem are performed between different datasets. It would be beneficial to see some analysis for inside dataset split, where the split is indicated by two groups of classes. This case is somehow similar to Cifar10 vs. Cifar100 comparison.

**Summary Of The Paper:**

The paper provides an interesting approach for OOD detection problem for flow models. The approach is based on random projections on the real line where KD statistical test is applied in order to compare two distributions. Two variants of the approach are considered 1 sample test where the comparison is made with respect to flow prior and 2 sample test where the comparison is made with respect to transformed samples from two datasets. The quality of the approach is compared agains standard reference methods.


**Summary Of The Review:**

I admire the simplicity of the approach and I tent to recommend to accept this work at this stage.

---

> ### Author Response · Authors · 2021-11-23
> **Response to Reviewr Kij3**
>
> Thank you very much for your positive comments, and we address your questions/concerns below:
>
> **Q1.** It seems that this framework is general and can be even applied to the bottleneck models like VAE or even for any low dimensional features extracted from the data. The question is can we apply this framework to any low dimensional representation of data or we need the invertible mapping that does not decrease the dimension and direct access to the likelihood.
>
> **Response:** Thank you for this comment.
> Indeed, our framework can be applied to bottleneck models with little modification. For invertible models, we gain additional advantages: (a) We do not lose any information in the latent space. In other words, the converse of Eq (3) also holds. In contrast, for a bottleneck model, it is possible that $T(X) \stackrel{d}{=} T(Y)$ but $X \stackrel{d}{\neq} Y$, incurring possibly false positives.
> (b) Many existing approaches are based on log-likelihood. Having an invertible model makes the comparison more fair.
>
> **Q2.** The second question is about Carleman’s condition. Considering the case where we use flow’s latent is it sufficient that it would be satisfied by Gaussian (flow latent distribution) or it should be satisfied by data distribution (the normalised Gaussian by determinant of the Jacobian)?
>
> **Response:** For detection in the latent space, it is sufficient that the latent distribution (e.g. Gaussian) satisfies Carleman's condition. For detection in the sample space, since real data (e.g. image pixels) are bounded, they always satisfy Carleman's condition.
>
> **Q3.** The model seems to be comparable to KLOD besides the CIFAR10 vs SVHN where it performs better. On the other hand, it has problems with CIFAR vs LeSUN setting. Do you have some intuition what is the reason and how to deal with that issue?
>
> **Response:** KLOD imposes strong assumptions on the latent distribution and the model accuracy (while we have no such assumptions), thus it is possible that KLOD may perform better if their assumptions happen to be met.
>
> For CIFAR vs LSUN, our experimental results indicate that this is a challenging case for our method. We examined the raw distributions of the two datasets, through per-dimensional mean and variance. The detailed result is given in the new Appendix G.8. In Figure 12, we verify that CIFARs and LSUN share a very close per-dimensional mean and variance, while for CIFARs vs. SVHN, although their per-dimensional mean share a similar range, their distributional shapes are different in terms of per-dimensional variances. This implies the CIFARs vs. LSUN pair is distributionally more similar and hence more challenging for our methods.
>
> The immediate solution is to increase the batch size. When batch size = 100, our methods (using RealNVP) can achieve AUROC of $0.97|1.00$ for CIFAR-10 vs. LSUN, and $0.90|0.99$ for CIFAR-100 vs. LSUN. When batch size is beyond 200, our methods can achieve perfect (both AUROC and AUPR reach 1.00) detection performance in both cases.
>
> **Q4.**  Most of the experiments that examine the OOD problem are performed between different datasets. It would be beneficial to see some analysis for inside dataset split, where the split is indicated by two groups of classes.
>
> **Response:**
> Thank you for this suggestion! In the revision, we've included additional experiments for the intra-dataset OOD detection on FMNIST and CIFAR-10, where one class is considered InD and the rest nine classes combined are considered OOD. Experimental results are presented in Table 9 in the new Appendix G.6. Compared to SOTA baselines, our method achieves the best intra-dataset OOD detection performance on FMNIST for all classes, while on CIFAR-10, our method yields near-best and more stable detection performance for all classes. Besides, there is no detection failure case (i.e., AUROC and AUPR below 0.5) in our method, which serves as another evidence of the effectiveness and generality of the proposed method.

---

> > ### Comment · Reviewer_Kij3 · 2021-11-30
> > **Thank you!**
> >
> > Thank you for your responses. I am entirely satisfied with your answers. Therefore, I am going to raise my score to (8: accept, good paper).

---

### Official Review · Reviewer_zvSD · 2021-11-04

**Correctness:** 4
**Technical Novelty And Significance:** 1
**Empirical Novelty And Significance:** 2
**Recommendation:** 8
**Confidence:** 4

**Main Review:**

## Pros

*Exploration of OOD in practice*:  While GOF-based OOD detection is theoretically well-founded, applying the technique successfully in practice for high-dimensional data is a challenge in its own right.  Thus, this paper provides value to practitioners, guiding them on how to make these crucial implementation choices.

*Ablations*: The paper also does a nice job of testing the efficacy of specific parts of the experimental pipeline---for example, autoencoders vs random projections, GOF vs two-sample tests.

## Cons

*Novel only in implementation*:  The method's novelty is restricted to the implementation details.  Ultimately, the core methodology is to apply the KS test to a reparameterized distribution, which is not an original contribution.  Applying a KS test to flows for OOD purposes is novel, to the best of my knowledge.  Moreover, the paper's other insights such as model quality vs OOD ability mostly back already supported points from earlier work (e.g. [Zhang et al, ICML 2021].

**Summary Of The Paper:**

This paper applies a goodness of fit test (KS test) to the latent space of normalizing flows for purposes of out-of-distribution detection.  To combat high-dimensionality and model misspecification, extensions such as random projections and two-sample tests are also proposed.  The results report AUROC on RNVP (batch size 5 and 10), comparing against a typicality test and a kl-based test.  The paper also tests using autoencoders vs rand projections, alternative divergences, effects of model misspecification, and detection in latent vs original feature space.  The findings are that the KS-test is indeed a practical choice for OOD detection via GOF testing.  Moreover, it is observed that better models don't necessarily mean better OOD detection, as described earlier by Zhang et al. [ICML 2021].

**Summary Of The Review:**

The paper is fairly well executed; however, I can't say that I learned much from the work as its contribution is mostly in the implementation details.  And it is unclear to what extent these details generalize beyond standard image benchmarks.

---

> ### Author Response · Authors · 2021-11-23
> **Response to Reviewer zvSD**
>
> Thank you very much for your  comments, and we address your questions/concerns below:
>
> **Q1.** *Novel only in implementation:* The method's novelty is restricted to the implementation details. Ultimately, the core methodology is to apply the KS test to a reparameterized distribution, which is not an original contribution. Applying a KS test to flows for OOD purposes is novel, to the best of my knowledge. Moreover, the paper's other insights such as model quality vs OOD ability mostly back already supported points from earlier work (e.g. [Zhang et al, ICML 2021].
>
> **Response:**
> Thank you for recognizing the novelty in our approach. We agree that there has been a surge of recent interests on understanding the applicability of generative models to OOD detection, and the results often boil down to different technical implementations. However, this does not mean implementation details are not important, and we believe our work brings a number of new insights in this important application:
> + A majority of existing works focus on likelihood based detection scores while our work proposes to combine random projection with sound statistical tests such as KS. It is important to conduct this investigation in order to delineate the role played by flow models from the detection algorithms.
> + We propose both the one-sample and two-sample tests, and investigate their respective pros and cons for well-estimated and under-estimated flow models. To our best knowledge, most existing approaches do not make this distinction while the relevant work of Zhang et al. (ICML 2021) only studied the effect of model mis-specification qualitatively.
> + Our method unify OOD detection in both the sample space and the latent space (and confirm the superiority of the latter approach on many datasets), and we do not make restrictive assumptions on the data and latent distributions. We believe the simplicity and generality of our approach is a strong advantage.
> + As confirmed by Reviewer XXZp, the observation that image generation quality does not necessarily correlate with OOD detection performance appears to be new, although we agree this further corroborates existing results in e.g. Zhang et al. (ICML 2021).
> Note that a priori it is not clear if image generation quality is correlated with likelihood either so our observation does not follow from existing ones. The point is that we now have (independently) verified the same surprising phenomenon, despite of employing a different detection algorithm that is not based on log-likelihood, on more datasets, flow models, and  metrics. Thus, we believe our results are still valuable and could help us further understand the pros and cons of flow models in OOD detection.
>
>
> **Q2.** It is unclear to what extent these details generalize beyond standard image benchmarks.
>
> **Response:** Thank you for this question. To check the performance of our algorithm on non-image datasets, in our modified submission we've included additional experimental results on time-series data, i.e. EEG and ECG (please refer to Table 10 in the new Appendix G.7 for details). We use a publicly available database (https://physionet.org/content/slpdb/1.0.0/) that collects multiple physiological signals from 16 subjects during sleep. We compared our results with two SOTA baselines in two settings, i.e. treating EEG as InD, ECG as OOD, and treating ECG as InD, EEG as OOD. The detection performance again indicates the generality and robustness of our method (especially GOD2KS).

---

### Author Response · Authors · 2021-11-23
**Overall major revisons**

We thank all reviewers for their careful reading of our paper and their suggestions. To address the concerns from reviewers, we have made the following main changes in our draft, colored in blue:
+ We modified our motivation for groupwise OOD detection in section 2.1, and added a synthetic experiment in Appendix E to illustrate (see Figure 7).
+ We re-run all the main experiments, and include AUPR as another evaluation metric in our results. Please refer to Tables 5, 6, 8. Our main claim remains unchanged, i.e. compared to SOTA baselines and GOD1KS, the proposed GOD2KS is more robust and generally satisfactory across all listed benchmark datasets. We also give more implementation details in Appendix B.
+ We modified all figure captions to make them more self-contained, e.g. Figures 3, 4, 5.
+ We added discussion on references that report related results to the correlation between model accuracy and OOD detection performance in Appendix G.5.
+ In Related works, we added discussion of more references that employ (partial) latent code for OOD detection with flow model and VAE, respectively.
+ We added Algorithm 2 for GOD2KS in Appendix B.
+ We conducted additional experiments on intra-dataset OOD detection. The results are summarized in Appendix G.6 (see Table 9). Compared to SOTA baselines, our method achieves the best intra-dataset OOD detection performance on FMNIST dataset for all classes, while on CIFAR-10 dataset, our method yields the near-best detection performance for all classes. Besides, there is no detection failure case in our method (metric below 0.5), which serves as another evidence of the effectiveness and generality of the proposed method.
+ We conducted additional experiments on detecting OOD in time-series data. In our experiment, we choose EEG and ECG, the two clinically important physiological signals, for analysis (see Table 10 and Appendix G.7 for details). Our method, especially GOD2KS, indicates more robustness against SOTA baselines. In the EEG/ECG setting, the typicality test fails, implying an overlap in density histograms of InD and OOD, and KLOD is slightly better than a random guess. In contrast, our GOD2KS achieves around 0.9 metrics in both experimental settings.
+ We added investigation on why CIFARs vs. LSUN setting is challenging for our method by examining the raw distribution through per-dimensional mean and variance. Please refer to Figure 12 and Appendix G.8 for details. Our experiments show that CIFARs vs. LSUN pair is indeed distributionally more similar and hence more challenging for our methods. The immediate solution is to increase the batch size. When batch size = 100, our methods (using RealNVP) can achieve AUROC of $0.97|1.00$ for CIFAR-10 vs.  LSUN, and $0.90|0.99$ for CIFAR-100 vs. LSUN. When batch size is beyond 200, our methods can achieve perfect (both AUC and AUPR reach 1.00) detection performance in both settings.
+ We conducted an additional experiment of including the average density as a straightforward baseline for comparison, and results are summarized in Appendix G.9 (see Table 11). It can be observed that directly averaging the density across a batch is not competitive.

---

### Public Comment · ~Dihong_Jiang1 · 2023-11-23
**Code link**

We forgot to mention that our code was released here last year: https://github.com/dihjiang/OOD-flow.

---

### Decision · Program_Chairs · 2022-01-20

**Decision:**

Accept (Poster)

**Comment:**

The paper investigates the use of flow models for out-of-distribution detection. The paper proposes to use a combination of random projections in the latent space of flow models and one-sample / two-sample statistical tests for detecting OOD inputs. The authors present results on image benchmarks as well as non-image benchmarks.

The reviewers found the approach well-motivated and appreciated the ablations. The authors did a good job of addressing reviewer concerns during the rebuttal. During the discussion phase, the consensus decision leaned towards acceptance. I recommend accept and encourage the reviewers to address any remaining concerns in the final version.

It might be worth discussing this paper in the related work: Density of States Estimation for Out-of-Distribution Detection https://arxiv.org/abs/2006.09273